# Balanced state of networks of winner-take-all units

Rich Pang  [1,2]*

**1** Princeton Neuroscience Institute, Princeton University, Princeton, New Jersey, United States of America, **2** Lewis-Sigler Institute for Integrative Genomics, Princeton University, Princeton, New Jersey, United States of America

* rp21@princeton.edu

**Data availability statement:** All code was written in Python 3 and is available at https://github.com/rkp8000/balanced-wta.

## Abstract

Irregularly timed action potentials, or spikes, are pervasively observed in the brain activity of awake mammals. However, the role of this temporal irregularity in neural computation is still not well understood. In canonical network models irregular spiking emerges via balanced, fluctuating input currents, leading to collective responses that track inputs linearly. How networks characterized by irregular spiking could support flexible nonlinear dynamics needed for general-purpose computation remains under ongoing debate. Here we characterize the dynamics of networks whose elementary unit is not a single neuron but a small group of neurons, with distinct tunings, that compete at each timestep via a winner-take-all (WTA) interaction. While WTA has long been proposed as an elementary functional motif in the brain and represents a powerful computational primitive, how large networks of such units behave has received less investigation. We show that these networks, like classic excitatory-inhibitory balanced networks, exhibit a chaotic fluctuation-driven regime characterized by sustained irregular activity resembling realistic cortical spiking, which we interpret as a multidimensional balance spread over several competing neural populations with different tunings. We develop a mean-field theory for the network, which shows how irregular spiking sustained by time-varying input fluctuations can support flexible nonlinear collective dynamics. Using the theory we predict and verify network regimes in which input fluctuations alone yield multistability, stable sequence generation, or complex heterogeneous firing rate dynamics—three core dynamical primitives thought to underlie memory-dependent neural computation—via consistent Poisson-like spiking produced through chaos. Thus, networks of WTA units support a chaotic fluctuation-driven regime characterized by irregular spiking that can power complex nonlinear collective dynamics. This represents a new model of brain activity capable of simultaneously reproducing realistic spike trains and diverse nonlinear firing rate patterns well posed for flexible computation, and which can be trained or fit to data.

Demos of spiking network simulations producing multistability and sequences can be run on a CPU and require about 8 Gb of RAM.

**Funding:** The author(s) received no specific funding for this work.

**Competing interests:** The authors have declared that no competing interests exist.

## Author summary

The irregular timing of action potentials, or "spikes," is one of the most pervasive features of neural activity in awake mammals, observed in a wide range of brain areas and conditions. However, the role of this striking temporal irregularity in neural information processing is still not understood. In network models, it has been particularly challenging to reconcile irregular spiking with flexible nonlinear "firing rate" dynamics, which are thought to be crucial for general-purpose neural computation. Here we present a network model, based on winner-take-all units akin to small groups of neurons, in which irregular spiking and flexible nonlinear dynamics go hand-in-hand and can be related via a concise mathematical theory. The result is a new model of brain activity capable of simultaneously capturing realistic spiking and firing rate dynamics, which is well posed for flexible computation and can be trained or fit to data.

## Introduction

Computation in the brain is thought to be orchestrated through the activity dynamics of populations of neurons, which respond to and transform inputs to process information [1–10]. Yet while this framework has provided transformative insights into neural computation, we still lack a comprehensive account of the role of the microscopic activity of the neurons involved, which in many brain areas and conditions manifests as sparse spike trains with high temporal irregularity and weak correlations [11–16]. This Poisson process-like spiking regime, termed the *asynchronous irregular* regime, has been most commonly observed in cerebral cortex of awake mammals, for example in primates performing working memory tasks [12,16], and has motivated substantial theoretical work investigating its origin and function [11,17–24]. Reconciling the asynchronous irregular regime with more complex, flexible collective dynamics in network models, however, has proved challenging [11]. Resolving this challenge will be crucial to understanding whether irregular spiking largely emerges from noisy biological processes, possibly co-opted for certain benefits (e.g. sampling input representations [25] or state transitions [26–28]), or if it plays a more central role in computation in general.

One core obstacle to resolving this problem is mathematical. At a cellular level irregular spiking is thought to arise via strongly fluctuating input currents [11,13,29,30], which foundational theoretical work showed could arise in strongly coupled random networks in which a balance of excitatory and inhibitory inputs emerges dynamically through network interactions—the classic *balanced state* [17–19]. In this model excitatory-inhibitory balance is central to irregular activity as excess inhibition or excitation would lead to quiescence or regular (non-Poissonian) spiking, respectively. (Note that excitatory-inhibitory balance can exist in networks that do not produce irregular spikes [31–33], and irregular spiking can emerge without balance, e.g. when driving neurons with noise; however, most spiking models of cortex assume that balance and irregular activity are tightly linked [11].) Strikingly, despite the spiking nonlinearity in the classic balanced state, collective responses—population-averaged firing rates—are linear in their inputs, converging to the form of $\mathbf{r} = A\mathbf{u}$ in the limit of large numbers of synaptic inputs, where $\mathbf{r}$ are the population firing rates, $\mathbf{u}$ is external input, and $A$ is a matrix [11,17–19]. This emergent linearity substantially limits the model's dynamical and computational abilities, as well as its ability to reproduce slow, complex, nonlinear dynamics observed in cortex [34–38]. Much work has thus sought to extend the classic balanced state to

nonlinear or slow computation, for example via finite-size theories [11,24,39], clustered connectivity [26,27], "semi-balance" from excess inhibition [23], or additional synaptic nonlinearities [40]. Nonetheless, how to construct robust network models in which flexible nonlinear dynamics co-exist with or potentially exploit irregular spiking remains a topic of ongoing debate [11,23,24,41–43].

Here we present an alternative to the classic balanced state that reveals how irregular activity can not only coexist with but directly support nonlinear firing rate dynamics. At the heart of the model is not a neuronal thresholding nonlinearity—by far the most common single-neuron nonlinearity used in spiking network models [17,19,42,44])—but instead a *multi-neuron* winner-take-all (WTA) nonlinearity. Specifically, we consider large networks of elementary units each corresponding to a small group of neurons with distinct tunings (external stimulus preferences), where within each unit the neurons compete with each other to activate at each time step. WTA has long been proposed as an elementary functional motif in the brain [45,46] and represents a powerful computational primitive [47], and WTA-like operations can be implemented quickly and efficiently in groups of model spiking neurons [48]. WTA operations also resemble the well-known computation of divisive normalization, in which neural activities are scaled by the summed activity of neurons with competing stimulus preferences [49]. Yet while the emergence of WTA competition or divisive normalization and their influence on sensory encoding have been studied extensively [48–55], how a multi-neuron WTA nonlinearity shapes global dynamics when repeated as a functional motif throughout a large network has received less attention. While fixed points of networks of WTA units have been examined [56], along with computational abilities of WTA units in specialized networks [57–59], the existence and properties of a more general and biologically realistic fluctuation-driven regime are not known.

Here we show that such a regime exists in networks of tuned WTA units. We find that in the high-gain or hard-WTA limit activity resembles realistic Poisson-like irregular spiking, and moreover supports flexible nonlinear firing rate dynamics. We refer to this dynamical regime as the balanced state of the network, in analogy with the classic excitatory-inhibitory balanced state [11,17–19]. In the network of WTA units, however, balance exists not between excitatory and inhibitory populations but among multiple populations of neurons with different tunings—hence the balance is "multi-dimensional." Individual WTA units receive effectively random, yet statistically balanced inputs from all active populations; but within a unit only one neuron can spike at each time step, resulting in Poisson-like activation patterns. Crucially, the multi-dimensional nature of this balance allows flexible, nonlinear collective dynamics to unfold simultaneous with and sustained by consistent Poisson-like spiking activity. Thus, networks of WTA units can support fluctuation-driven irregular activity capable of powering flexible nonlinear collective dynamics.

## Model

To approach this problem theoretically we consider an idealized network of $N$ elementary "units," each corresponding to a group of $D$ neurons. Within each unit we let each neuron be "tuned" to a discrete external input $u_d \in \{u_1, \dots, u_D\}$, with each unit containing exactly one neuron per tuning; hence we can label any neuron in any unit by its tuning $d \in \{1, \dots, D\}$. For instance, a tuning $d$ might represent a neuron's preferred stimulus orientation [60,61]. For simplicity we assume there is no overlap in the tunings, i.e. each neuron is tuned to exactly one of the $D$ external inputs.

## Terminology

Each group of $D$ neurons we refer to as a *unit*. All $N$ neurons with the same tuning $d$ we refer to as a *co-tuned population*, or simply *population*. Thus, in total the network contains $N \times D$ neurons, and there are $D$ different tunings or populations.

Note also that we will refer to the high-gain limit of the network, in which each neuron is either "on" or "off," as *spiking*. This contrasts with more common definitions of spiking networks, which usually connote leaky integrate-and-fire neurons that operate in continuous time and integrate their inputs over a finite timescale (the membrane time constant), emitting spikes when the integrated input crosses a spiking threshold [19]. For simplicity the network we study operates in discrete time, but intuitively one can think of each "on" event as corresponding to a spike or action potential, and the timestep of the update rule as being on the order of a typical neuronal membrane time constant (∼10–20 ms [62]).

Our study also involves two classes of dynamical systems, which we will frequently compare. The first class (Eq 1–4) describes the "microscopic" activity dynamics of each neuron in each WTA unit, which we will typically refer to as the *full* or *full spiking* (in the high-gain limit) dynamics. The second class (Eq 5–10) directly describes the dynamics of the "macroscopic" firing rates $\mathbf{r}^t \in \mathbb{R}^D$ of the $D$ populations, where the firing rate of a population is the average spiking activity of the $N$ neurons in that population at time $t$—these dynamics are derived as a mean-field description of the full spiking dynamics, and which we will refer to as the *mean-field* dynamics for $\mathbf{r}^t$.

## Update rule

Let the activation state of unit $i$ at time $t$ be given by a vector $\mathbf{y}_i^t \in [0, 1]^D$, where $y_{id}^t$ is the activity level of neuron $d$ in the unit. Within each unit we let the $D$ neurons "compete" with each other in a WTA-like fashion, such that $\sum_d y_{id}^t = 1$ for all $t$. This also means that every unit is always active—the only thing that changes is *which* neuron(s) are active at each timestep. This is supported by neural population recordings in monkey prefrontal cortex and monkey and mouse visual cortex, where, e.g., the mean or total activity across the population remains close to constant across task epochs even though individual neurons may differ [38,63].

We implement the WTA competition within each unit via a Softmax function. Given input $\mathbf{x}_i^t \equiv [x_{i1}^t, \cdots, x_{iD}^t] \in \mathbb{R}^D$, the activation of unit $i$ at time $t$ is given by

$$\mathbf{y}_i^t = \text{Softmax}(g\mathbf{x}_i^t) \tag{1}$$

where

$$\left(\text{Softmax}(g\mathbf{x}_i^t)\right)_d = \frac{\exp(gx_{id}^t)}{\sum_{d'=1}^D \exp(gx_{id'}^t)}, \tag{2}$$

with the parameter $g$ acting as a gain term. In the limit $g \to \infty$ the Softmax function becomes a hard WTA nonlinearity, such that $\mathbf{y}_i^t$ becomes a 1-hot vector indicating which of the $D$ neurons in unit $i$ received the largest input at $t$.

Inputs to neurons arise through network interactions (Fig 1a). Specifically, the inputs $\mathbf{x}_i^t$ to unit $i$ depend on the previous network state through

$$x_{id}^t = \sum_{d'=1}^D \sum_{j=1}^N J_{ij}^{dd'} y_{jd'}^{t-1} + u_{id}^t \tag{3}$$

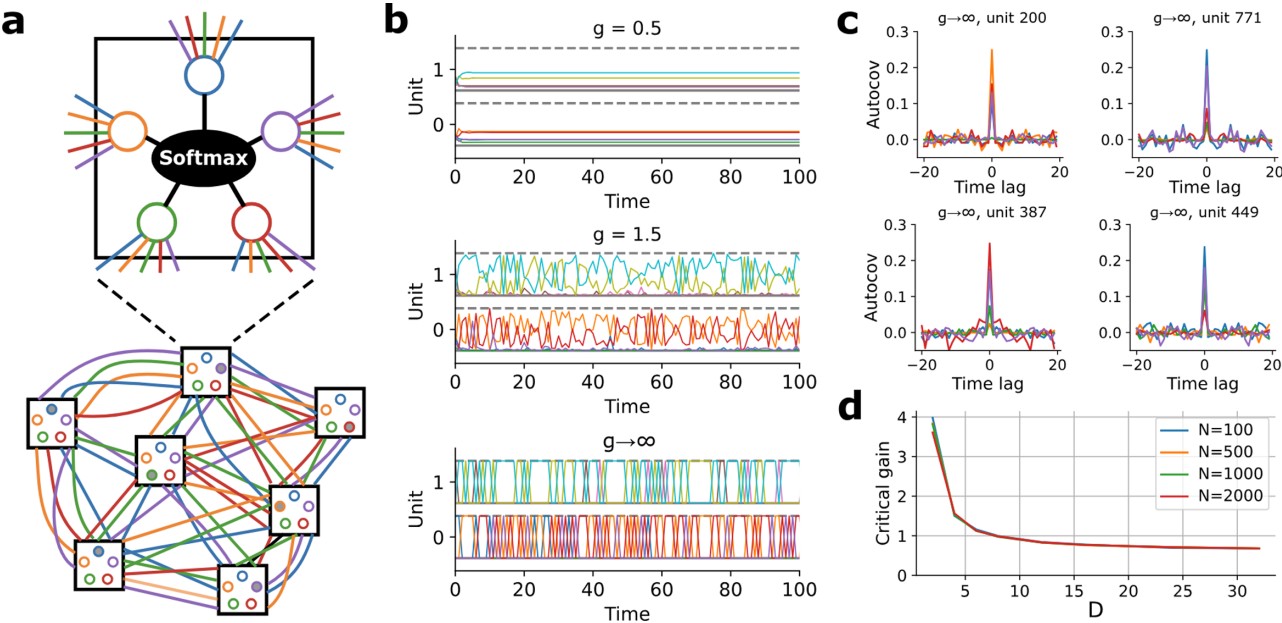

**Fig 1. Transition to chaos in network of WTA units.** (a) Network schematic. Each unit contains $D$ neurons that compete at each timestep through a Softmax nonlinearity. Inputs to each neuron in each unit can come from any other neuron in any other unit in the network. (b) Example activity dynamics of two example units (color = tuning) for three different gain levels (top to bottom). Dashed gray line indicates activity level of 1. (c) Numerically estimated autocovariance functions of 4 units in a network with infinite gain (N=1000, D=5). (d) Critical gain (estimated by identifying when a near-infinitesimal perturbation switched from being contracted to being expanded by the network dynamics) vs $D$ and $N$.

where $J_{ij}^{dd'}$ is the weight from neuron $d'$ in unit $j$ onto neuron $d$ in unit $i$, and $u_{id}^t$ is external input. In general, a single neuron can receive inputs from any neuron in any population. Equations 1–3 specify the complete update rule for the model. Unless stated otherwise, there is no noise—all variability in activity arises through deterministic network interactions.

Note that the total activity $\sum_{i,d} y_{id}^t = N$ for all $t$, which in the $g \to \infty$ limit corresponds to exactly $N$ neurons spiking at each timestep. Constant WTA-like activity, as a functional consequence of inhibitory feedback, was also proposed in an earlier "synfire chain" model of cortical dynamics [64–66], and is supported by neural recordings in which mean activity across the total recorded population is constant across task epochs, even under stimulation [38,63]. This stability of total activity also means the network is spontaneously active even without external input.

## Network weights

The network weights are given by a 4-tensor $J_{ij}^{dd'}$, whose elements we take to be i.i.d. (independent and identically distributed) Gaussian, except with tuning-dependent means and variances. Specifically, we let

$$J_{ij}^{dd'} \sim \mathcal{N}\left(\mu_J(d,d')\frac{D}{N}, \sigma_J^2(d,d')\frac{D}{N}\right) \tag{4}$$

where $\mu_J(d, d')$ and $\sigma_J^2(d, d')$ specify the means and variances of synaptic weights from neurons with tuning $d'$ onto neurons with tuning $d$. As we will see, the $1/N$ scaling of the variances promotes fluctuation-driven dynamics. Note also that we often refer to $\sigma_J(d, d')$, which is just the square root of $\sigma_J^2(d, d')$.

# Results

## Unstructured network

To study the baseline behavior of the network we first investigate the case of unstructured connectivity, in which $\mu_J(d, d') = 0$ and $\sigma_J^2(d, d') = 1$ are independent of pre- and post-synaptic tuning, and absent external input ($u_{id}^t = 0$). For sufficiently low gain (e.g. $g = 0.5$), the network quickly evolves to a fixed point, reflected in the constant activity levels of individual units (Fig 1b, top). As $g$ is increased, however, autonomous temporal fluctuations emerge (Fig 1b, middle), and in the $g \to \infty$ limit individual neurons produce sustained highly irregular switches between on and off states (Fig 1b, bottom). The temporal irregularity of single-neuron activity in the $g \to \infty$ limit is quantified in the autocovariance functions estimated from numerical simulations (Fig 1c). Estimating (numerically) whether microscopic perturbations (swapping the activity levels of two neurons within one unit) expand or contract suggests that the network exhibits a transition to chaos at a critical gain level that decreases with $D$, and as $D$ grows becomes independent of $N$ (Fig 1d; presumably for sufficiently large $N$). Note that strictly speaking for finite $N$ the $g \to \infty$ limit is not chaotic, since the dynamics unfold deterministically on a discrete space so must eventually fall into a periodic orbit; however, they are effectively chaotic, as measured by sensitivity to small perturbations (S1 Fig), and we expect them to approximate true chaos more closely as $N \to \infty$ [22,67–69]. Thus, similar to transitions to chaos in other random networks [1,22,70], the network of WTA units becomes chaotic above a critical gain, producing activity patterns suggestive of the irregular activity patterns observed in single neurons in the brain.

## Spiking network representation

In the $g \to \infty$ limit, the model admits a representation as a spiking neural network. At each time $t$ each neuron is either on ($y_{id}^t = 1$) or off ($y_{id}^t = 0$), allowing us to treat the collective dynamics as population spike trains. A raster plot of a subset of neurons in such a network reveals an irregular spiking pattern (Fig 2a) strongly resembling the well-known asynchronous irregular state of cortical firing [11,13–15], and visually similar to activity produced by classic balanced networks [17–19]. Numerical estimates reveal approximately exponential single-cell inter-spike interval (ISI) distributions (Fig 2b), a key signature of spikes emitted from a homogeneous Poisson process. A concise measure of how similar a spike train is to a Poisson process is the ISI coefficient of variation (ISI CV), or its non-stationary equivalent ISI $CV_2$ [14,16] (Methods), both of which are near unity for a homogeneous Poisson spiking process; ISI $CV_2$ (but not ISI CV) will also be near unity for an inhomogeneous Poisson process. Indeed, the ISI $CV_2$ distributions in the unstructured network are near unity (Fig 2c), recapitulating a key signature of irregular spiking in the brain [15,16]. The network also produces a right-skewed distribution of time-averaged firing rates across each population (Fig 2d), resembling firing rate distributions observed in biological spike trains [71]. Finally, the mean firing rate decreases as $1/D$, suggesting no lower limit on baseline firing rate (Fig 2e).

Thus, in the high-gain limit the network produces activity equivalent to irregular spiking, recapitulating a fundamental dynamical feature of realistic neural activity, and with no apparent minimum firing rate. In analogy to the well-studied excitatory-inhibitory balanced

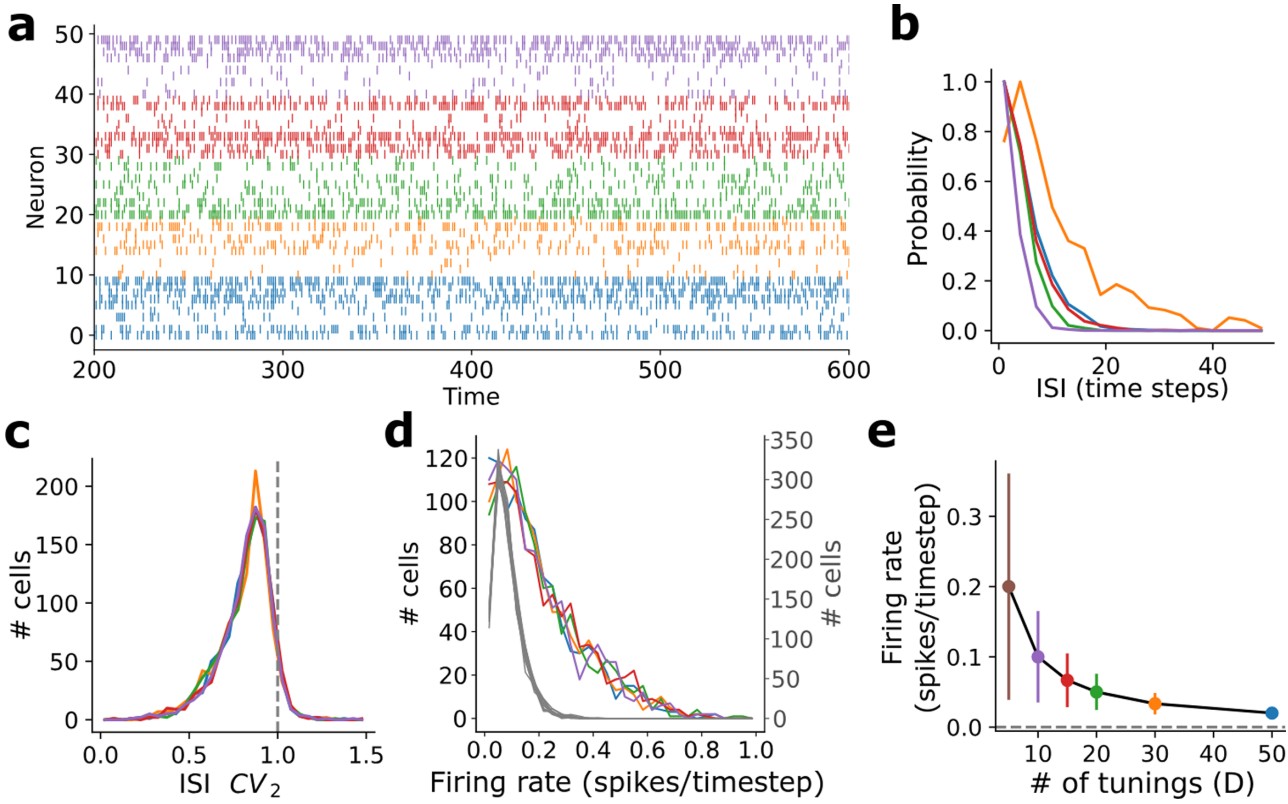

**Fig 2. Spiking network representation of hard winner-take-all limit.** (**a**) Raster plot for one network simulation with $D = 5$, showing a few example neurons with each tuning (color = tuning). (**b**) Interspike interval (ISI) distribution for one example unit from each tuning. (**c**) Distribution of (non-stationary) inter-spike interval coefficient of variation ISI $CV_2$ for each co-tuned population. (**d**) Distribution of time-averaged single-neuron firing rates (number of spikes divided by simulation length [3000]) for each co-tuned population (colored curves, $D = 5$). Gray curves show equivalent for a network with $D = 12$ different tunings (not colored). (**e**) Population firing rates vs $D$ for a network with 1000 units. Firing rates are computed as averages across neurons within each co-tuned population. Solid line shows theory ($1/D$); scatter points and error bars show simulation results. Error bars are standard deviations over tunings and time.

state [11,17–19], the multi-dimensional balance of inputs from different co-tuned populations unlocks a fluctuation-driven dynamical regime in which input fluctuations perpetuate themselves through irregular spiking.

## Theory

To understand the capacity of this network to generate computationally beneficial collective dynamics we develop a mean-field theory, in the $g \to \infty$ limit, for the time evolution of the population firing rates. To this end, although the true evolution of the neural activity is deterministic Eq. 1–3, we assume it is sufficiently chaotic to be described statistically. Specifically, we treat the activity probability distribution of a single, "typical" representative unit $i^*$ as statistically representing the collective state. We are then interested in the time evolution of $\mathbf{P}^t \in \mathbb{R}_+^D$, the categorical distribution over which of the $D$ neurons in $i^*$ spikes at $t$. We can define the population-averaged *firing rates* of the $D$ co-tuned populations as $\mathbf{r}^t \equiv \mathbf{P}^t/\Delta t$, with $r_d^t$ the expected number of spikes per $\Delta t$ of a neuron in population $d$ at time $t$ (assuming a neuron can spike at most once per $\Delta t$). For simplicity we choose units where $\Delta t = 1$ (reflecting the 1-timestep update rule of Eq. 1–3) so that $\mathbf{r}^t = \mathbf{P}^t$.

As the firing rate vector $\mathbf{r}^t$ is equivalent to a discrete probability distribution, it evolves on the regular (D-1)-simplex $\mathcal{S}_D$ (which has $D$ corners) (Fig 3a and 3b). Accordingly, it can be shown (S1 Appendix) that the firing rates $\mathbf{r}^t$ evolve according to:

$$r_d^t \approx \int_x \mathcal{N}\left(x; \mu_d, \sigma_d^2\right) \prod_{d' \neq d} \Phi\left(x; \mu_{d'}, \sigma_{d'}^2\right) dx \qquad (5)$$

where $\Phi$ is the cumulative normal density and $\{\mu_d\}$ and $\{\sigma_d^2\}$ are the input means and variances to neurons with different tunings, which depend on the previous firing rate vector $\mathbf{r}^{t-1}$. Intuitively, Eq 5 gives the probability that a sample from the Gaussian with the $d$-th mean and variance is larger than samples from all the others, reflecting the WTA competition. Note that the theory does not incorporate the heterogeneity of firing rates within each population (Fig 2d), which may account for some of the small errors in predicting the full network simulations.

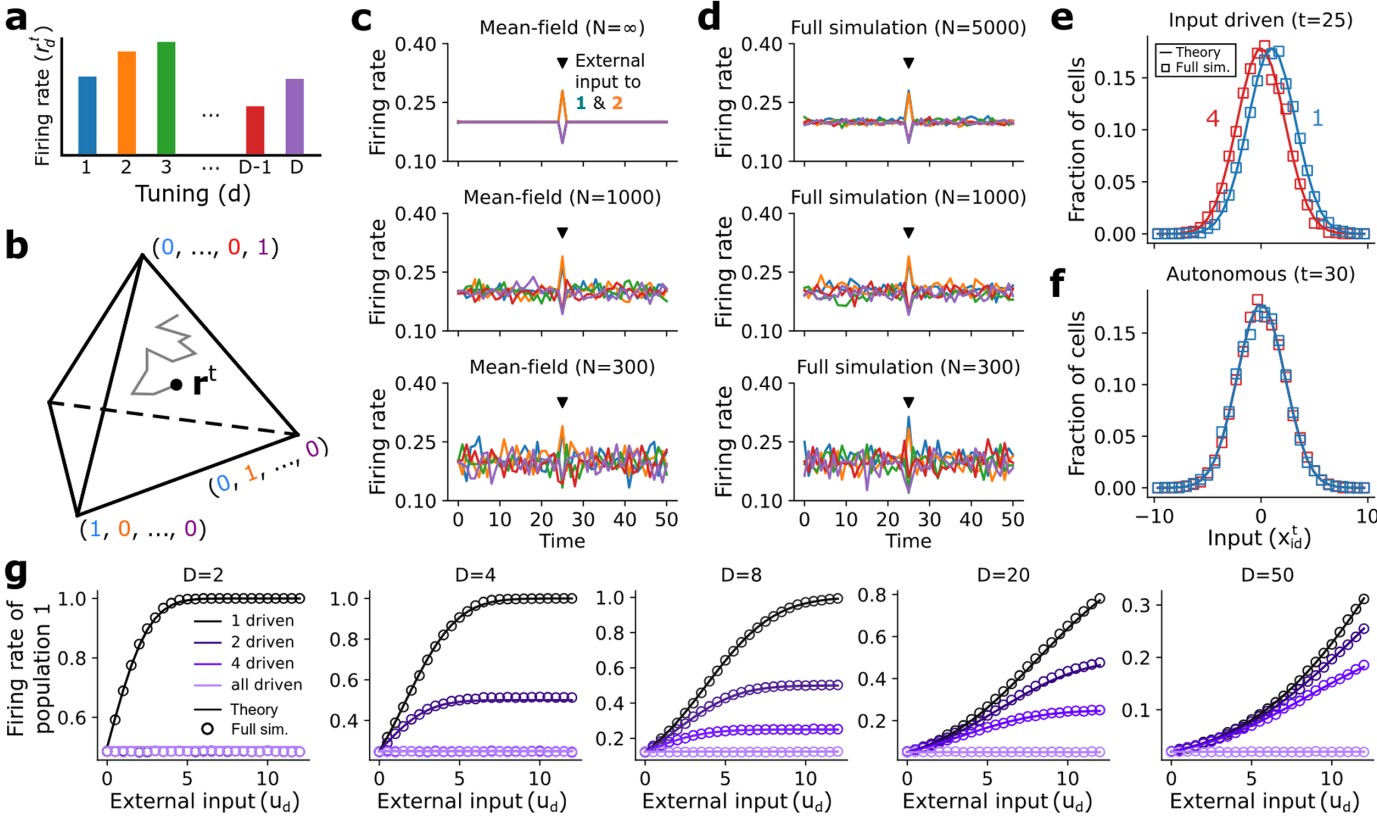

**Fig 3. Mean field theory and input response.** (**a**) Schematic of the mean-field variable (the population firing rates $\mathbf{r}^t$), which is equivalent to a categorical probability distribution $\mathbf{P}^t$ over which neuron (defined by its tuning) is active in the representative unit $i^*$ at time $t$. (**b**) $\mathbf{r}^t$ evolves on a (D-1)-dimensional simplex $\mathcal{S}_D$ (which has $D$ corners). (**c**) Simulations of mean-field dynamics for unstructured networks ($D = 5$) of three different sizes, with a 1-timestep external input provided to tunings 1 and 2 at $t = 25$. (**d**) Full network simulations corresponding to (C) (except that the large network uses $N = 5000$). (**e**) Mean-field predictions and full simulation ($N = 5000$) estimations of input distributions to neurons with tunings 1 and 4 at the time when the external input is applied. (**f**) As in **e** except at a time when no external input is applied. (**g**) Time-averaged response of the first co-tuned population to constant external input applied to population 1, populations 1 and 2, populations 1–4, or all co-tuned populations. Circles give averages across three initial conditions of the full network simulation ($N = 1000$).

The theoretical input means and variances are given by

$$\mu_d \approx D \sum_{d'} \mu_J(d,d') \frac{n_{d'}^{t-1}}{N} + u_d^t \tag{6}$$

and

$$\sigma_d^2 \approx D \sum_{d'} \sigma_J^2(d,d') \frac{n_{d'}^{t-1}}{N} + v_d^t \tag{7}$$

where $u_d^t$ and $v_d^t$ are the mean and variance of the external input to neurons with tuning $d$, and $n_{d'}^{t-1}$ is the number of neurons with tuning $d'$ that spiked at $t-1$. For finite networks the latter is effectively stochastic:

$$\mathbf{n}^t \equiv [n_1^t, \dots n_D^t] \sim \text{Multinomial}(\mathbf{r}^t, N). \tag{8}$$

Thus, one expects fluctuations in the population firing rates even when the firing rates at the previous timestep are known exactly. For large networks, however, the input means and variances approach

$$\mu_d \approx F_\mu \left[ \mathbf{r}^{t-1} \right] \equiv D \sum_{d'} \mu_J(d,d') r_{d'}^{t-1} + u_d^t \tag{9}$$

and

$$\sigma_d^2 \approx F_\sigma \left[ \mathbf{r}^{t-1} \right] \equiv D \sum_{d'} \sigma_J^2(d,d') r_{d'}^{t-1} + v_d^t \tag{10}$$

such that the time-evolution of $\mathbf{r}^t$ becomes deterministic, even when the underlying spiking dynamics remain irregular. Thus, in total, the mean-field equations for the firing rate dynamics are parameterized by $D$, $N$ (for finite-size simulations), and two $D \times D$ matrices $\mu_J$ and $\sigma_J^2$. In contrast to the linear steady-state equations of the classic balanced state [17,18], the mean-field equations governing our network are generically nonlinear with respect to both external inputs and the previous network state, even when $N \to \infty$. This suggests the network may be able to exhibit more expressive collective dynamics than a linear system alone.

## Input response

Simulating the mean-field dynamics (Eq 5–10) lets us predict the behavior and input responses of the full spiking network (Eq. 1–4). For instance, the theory predicts that in infinitely large unstructured networks spontaneous activity will be evenly balanced across the $D$ populations and perturbations immediately corrected (Fig 3c), which is recapitulated in full simulations of large networks (Fig 3d, $N = 5000$). Notably, the theory also lets us predict the dynamics of finite networks (Eq. 6–7), which similarly balance activity across all populations and immediately correct the perturbation (Fig 3c), but atop a blanket of noise that decreases with $N$, which is reflected in the full simulations with matched $N$ (Fig 3d). Eq 9 and Eq 10 also give an exact description of the (Gaussian) input distribution when the previous state and external input mean and variance are known, which is corroborated by the full simulation (Fig 3e and 3f).

We can also predict the response of the unstructured network to constant external input, a common experimental stimulus [72]. For simplicity we set the external input variances $v_d^t = 0$. In general, the network response depends on the total number of co-tuned populations in the network along with how many are driven by the external input (Fig 3g). In all cases, the WTA

nonlinearity (Eq. 1) bounds the total firing rate of any population at 1 (since the maximum rate is one spike per timestep). When $K$ out of $D$ populations are driven, the firing rate of each individual population saturates at $1/K$, and as expected the network shows no response when all populations are driven (since this just shifts the baseline atop which the WTA competition takes place). When only a few out of a large number of populations are driven, however, the driven populations exhibit an additional supralinear response at weak inputs (Fig 3g, right), before eventually reaching the sublinear saturating regime at strong inputs.

One feature of this system is that neurons' responses to their preferred stimulus are suppressed in the presence of non-preferred stimuli. In Fig 3g (D=4), for instance, population 1 exhibits a strong response when driven alone but a weaker response when population 2 is also driven. This is similar to the phenomenon of cross-orientation suppression observed in visual cortex, in which presentation of a non-preferred orientation suppresses a neuron's response to its preferred orientation [34]. Intuitively, this reflects the within-unit WTA competition, which influences the population response. This behavior cannot emerge in the classic balanced state, which responds linearly to external inputs [17,18], although nonlinear responses can emerge if populations are allowed to be silenced via excess inhibition ("semi-balance") [23]. The input responses of our network are instead closer to those of networks operating in the "loose balance" regime, which also exhibits cross-orientation suppression [11,39], except here our network's response is computed in the limit of an infinite number of pre-synaptic inputs.

## Stability of the uniform distribution

The return to evenly distributed activity (across all populations) after external input is removed (Fig 3c and 3d) can be understood using our theory. In unstructured networks, where $\mu_J(d, d') = 0$ and $\sigma_J^2(d, d') = 1$, we see that absent external inputs the uniform distribution is stable. This is because $\mu_d = 0$ (Eq 9), and from Eq 10:

$$\sigma_d^2 = D \sum_{d'} r_{d'}^{t-1} = D \quad \forall d. \tag{11}$$

Thus, $\mathbf{r}^t$ becomes the distribution over the argmax of samples from $D$ identical Gaussians (Eq 5), which by symmetry must be uniform, so $\mathbf{r}^t = \mathbf{1}/D$. Therefore all network states are attracted back to the uniform fixed point in a single time step. Stability does not depend on $D$, hence uniform distributions over a large number of tunings are also stable, even though individual neurons spike at a mean rate of $1/D$. In agreement with simulation, this means firing rates can be made arbitrarily low, yet nonzero, simply by increasing $D$ (Fig 2e).

## Structured networks

More interesting dynamics arise if we introduce structure into the network connectivity. To this end we now explore network implementations of three core dynamical primitives thought to support memory-dependent neural computation in the brain: (1) multistability, which allows information to persist over time in macroscopic attractor states of the network [73,74]; (2) sequence generation, in which activity evolves along a 1-dimensional manifold, which is thought to support decision-making and time-keeping [4,75,76]; and (3) slow (relative to the single-neuron response timescale), complex, nonlinear heterogeneous firing rate dynamics, which are thought to facilitate learning of flexible nonlinear temporal computations and motor outputs [2,77].

Notably, our theory lets us predict how such macroscopic dynamics could emerge via multiple different network mechanisms. First, we can adjust only the mean weights $\mu_J(d, d')$ among the different populations. Alternatively, we can adjust only the weight variances $\sigma_J^2(d, d')$—in this scenario all mean inputs are always zero (Eq 6, Eq 9), hence macroscopic dynamics can only emerge from changes in the input variances over time, i.e. the dynamics are forced to be *fluctuation-driven*, which promotes irregular spiking. In general, one can also adjust combinations of the weight means and variances. Below we demonstrate advantages of fluctuation-driven over mean-driven implementations of both multistability and sequence generation, then show how the fluctuation-driven regime alone can support complex, nonlinear firing rate dynamics mediated by irregular spiking. In each setting, we first study the mean-field dynamics (Eq 5–10) of the $D$-dimensional vector $\mathbf{r}_t$ of firing rates of the $D$ populations, then simulate the corresponding full spiking network (Eq 1–4) governing the dynamics of the single-neuron activities $\{\mathbf{y}_i^t\}$.

## Multistability

A long-standing hypothesis for the basis of slow neural dynamics supporting working memory is multistability of the network state through persistent spiking [12,16,26,27,78,79]. As multistability does not emerge in randomly coupled balanced spiking networks of excitatory and inhibitory neurons without fine-tuning [17,80,81], how to achieve multistability in networks that exhibit irregular spiking in all stable states remains an open question. In particular, introducing positive excitatory feedback among neurons (e.g. in clustered balanced networks) produces multistability, but spiking becomes less irregular [26,27], in contrast with experiments. Multistability can also emerge in semi-balanced networks, in which certain populations are allowed to be silent [23]. Alternatively, one study showed how certain synaptic nonlinearities could produce two stable states characterized by different input variances [40], recapitulating the near-unity ISI CVs observed across task epochs in cortex [16]; while this model required additional synaptic nonlinearities via short-term plasticity rules, it demonstrates the consistency of irregular spiking with nonlinear global dynamics flowing through input variances.

**Mean-driven multistability.** To test our network's ability to exhibit multistability, we first explored "mean-driven" multistability. To do so we increased the mean weights within each co-tuned population (S2 Fig). This reflects traditional working memory models, in which network states are stabilized by positive feedback, but typically at the cost of less irregular hence less realistic spiking [26,27,73,82]. This was also reflected in our model—as we increased the diagonal of the mean weights $\mu_J(d, d) = (\mu_J)_0$, our theory predicted a sharp transition from monostability (of the uniform distribution) to exactly one of the populations being maximally active and the rest silent (S2 Fig). The theoretical predictions were recapitulated in the full simulations (S2 Fig), which additionally revealed the consequent regularity of spiking, which is inconsistent with experimental observations [16].

Using our theory we were able to explain the mono-to-multistable transition of the mean-driven network mathematically. In our model multistability arises when the uniform distribution ($\mathbf{r}^t = \mathbf{1}/D$) destabilizes, which forces the macroscopic state elsewhere, in the mean-driven case to a corner of the simplex on which the dynamics unfold (Fig 3b). Our theory let us predict when this would occur, which corresponds to the maximum eigenvalue $\lambda_{max}$ of the Jacobian of the mean-field dynamics (Eq 5), evaluated at $\mathbf{1}/D$, exceeding 1. This occurred when $(\mu_J)_0$ was sufficiently large and when $D$ was sufficiently small (S3 Fig)—increasing $D$ in a multistable network can make the network monostable, as there are more co-tuned populations to compete with. This in turn let us predict the expected number of active populations,

along with the information content of the network state ($\log_2$ of the number of active populations), which was approximately recapitulated in the full network simulations (S3 Fig). Because in the mean-driven multistable network only 1 population was active at a time, the information content of a stable state is bounded at $\log_2 D$ bits.

**Fluctuation-driven multistability.** Next, we explored the possibility of fluctuation-driven multistability in our network. To this end, we attempted to induce multistability by increasing the diagonal of the weight *variances*, $\sigma_J^2(d, d) = (\sigma_J^2)_0$ (Fig 4a), with all off-diagonal weight variances set to $\sigma_J^2(d, d' \neq d) = (\sigma_J^2)_1$, which led to substantially different macroscopic behavior from the mean-driven case. Our theory predicted that, in contrast to the mean-driven regime, the fluctuation-driven regime should exhibit a much smoother transition to multistability in which not one but several populations are active and the rest quiescent (but not necessarily silent) (Fig 4b), which was recapitulated in full simulations that also suggested the presence of highly irregular spiking (Fig 4c). Thus, the network can produce multistability in a fluctuation-driven regime as well.

Computing $\lambda_{max}$ for the fluctuation-driven case also revealed a transition to multistability (via destabilization of the uniform distribution) given sufficient feedback (here through

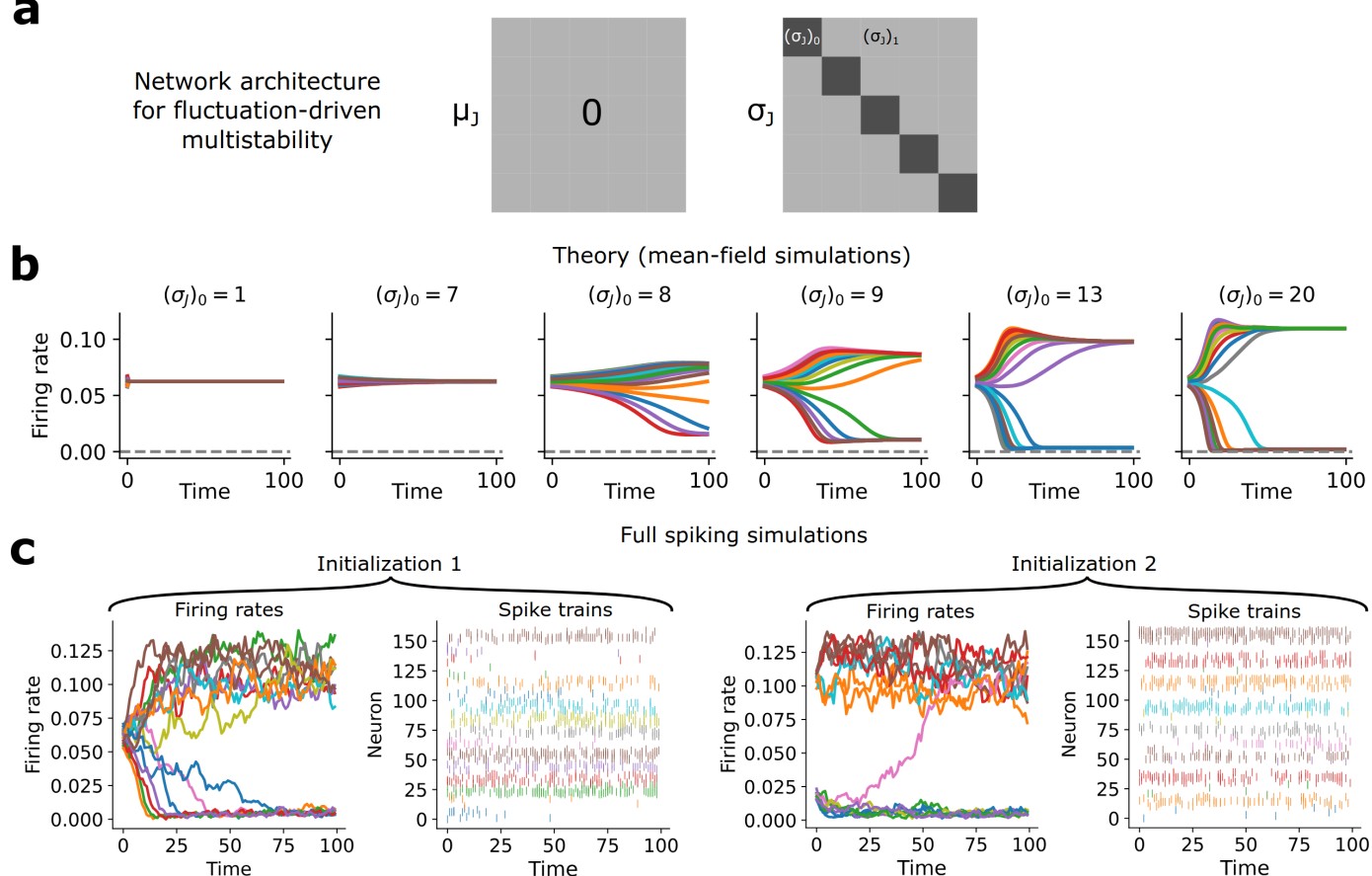

**Fig 4. Fluctuation-driven multistability.** (**a**) Schematic of network parameters (means and variances). (**b**) Mean-field simulations ($N \to \infty$) of firing rate dynamics for 6 different $(\sigma_J)_0$. (**c**) Example firing rate dynamics and spike trains from a full simulation of the fluctuation-driven multistable network ($(\sigma_J)_0 = 13$, $N = 3000$, $D = 16$).

increased weight variances) (Fig 5a). Unlike the mean-driven case (S3 Fig), however, multistability in the fluctuation-driven network emerges only when $D$ is sufficiently large, after which a subset of the $D$ populations remain active in each stable state. Because multiple populations are active in each network state, similar to the clustered balanced networks of [26, 27], the network can hold much more information than if only one population were active (Fig 5b). Curiously, in the $N \to \infty$ limit when $(\sigma_J^2)_0 \gg (\sigma_J^2)_1$, the uniform distribution destabilizes when $D = 11$, leading to exactly 10 active populations for $D \geq 10$. The number of populations at which this transition occurs, however, can be controlled by changing the diagonal $(\mu_J)_0$ or off-diagonal $(\mu_J)_1$ mean weights (Fig 5c and 5d), such that the information capacity of the network is not limited by only 10 populations being active. In general, the number of active populations increases as $(\mu_J)_0$ decreases (Fig 5c). Further, when more populations are active the firing rate of neurons in those populations decreases, suggesting stable states where neurons are active at low firing rates.

The fluctuation-driven multistable network behaves very differently from the mean-driven network. For instance, examining the fluctuation-driven simulations more carefully revealed that different stable states are distinguished by different input variances, rather than means (Fig 5e). Moreover, spiking was highly irregular (near-unity ISI CV and ISI CV$_2$) in both the

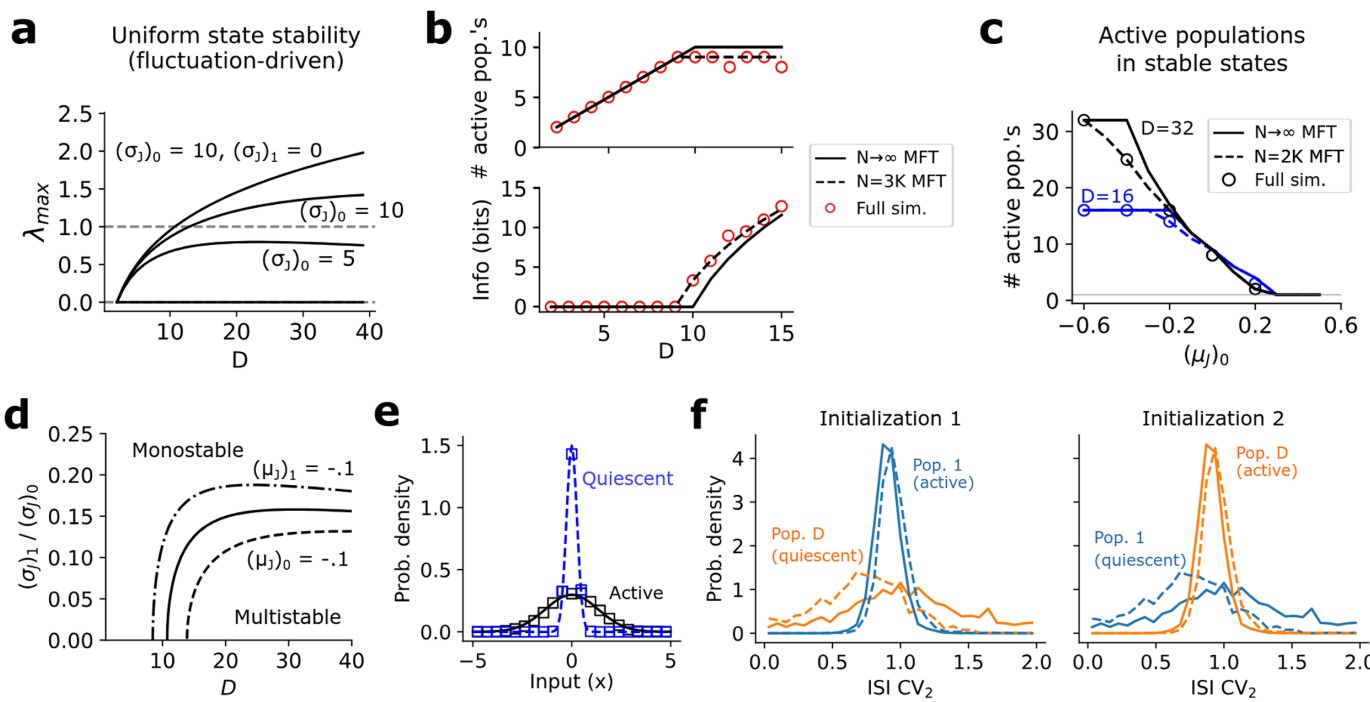

**Fig 5. Properties of fluctuation-driven multistable network.** (**a**) Maximum eigenvalue $\lambda_{max}$ of the Jacobian of the $N \to \infty$ mean-field dynamics evaluated at the uniform distribution $\mathbf{1}/D$, as a function of $D$. $(\sigma_J)_1 = 1$ for the bottom two traces. (**b**) Number of active populations (top) or equivalently information content (bottom) vs $D$ $((\sigma_J)_0 = 13)$. Solid lines are theory; circles are full simulation results ($N = 3000$). (**c**) Theoretical predictions of the number of active populations vs the diagonal term $(\mu_J)_0$, when $(\sigma_J)_0 = 1$, $(\sigma_J)_1 = 0$, with circles depicting median over 15 initializations ($N = 2000$). (**d**) Theoretical phase portrait depicting when the network is multistable. (**e**) Theoretical (lines) and simulation predictions (squares) of input distributions to active and quiescent populations in the fluctuation-driven multistable network. (**f**) ISI CV$_2$ distributions for two co-tuned populations using extended runs (1500 timesteps) of the fluctuation-driven multistable simulations in Fig 4c. Dashed traces show ISI CV.

active and quiescent states of a population (Fig 5f), which was also observed in monkey pre-frontal cortex during a working memory task [16], with irregular spiking in the model emerging through chaos (S4 Fig). Thus, our network admits a fluctuation-driven multistable regime in which spiking is consistently irregular regardless of the network state, suggesting a new solution to multistability in a brain-like dynamical regime. In our model these phenomena emerge ultimately through the multi-dimensional WTA nonlinearity and collective interactions of the neurons, without introducing any other single-neuron or synaptic timescales beyond the 1-timestep update rule (Eq 1–3). This also suggests a fundamentally different form of multistability from traditional models, in which activity is stabilized via positive excitatory feedback [73,78,80,83–85]. Here activity is stabilized via increased input variances (similar to [40], except without requiring additional synaptic nonlinearities), corresponding to co-tuned neurons with higher variance, rather than higher mean, recurrent connection weights.

## Sequence generation

A second common dynamical motif thought to serve memory- and time-dependent computation in the brain is the generation of stereotyped neural activity sequences [4,75,76,86,87]. Unlike multistability, sequence generation allows information to be not only held but transformed over time. To assess our network's capacity to generate sequences we destabilized the multistable network, in a similar spirit as [87]. First we placed each population on a ring, such that the $D$ populations were evenly spaced between $-\pi$ and $\pi$ (we focus on periodic sequences). Starting with either the mean- or fluctuation-driven multistable network we then introduced an asymmetric change to the network connectivity, corresponding to unidirectional increases in the weights (either in the mean or variance) from each population to its clockwise neighbor.

**Mean-driven sequence generation.** As with the multistable network, we first explored the consequences of unidirectionally increasing the mean weights $(\mu_J)_2$ linking neighboring populations (S5 Fig). Similar to the multistable case, our theory predicts a transition from the network producing no sequential activity (one population saturated and the rest silent) to producing very fast sequences, with one population approximately saturated at each time point (S5 Fig). This was corroborated in the full simulations, where fast sequential firing rate dynamics were accompanied by regular spike trains (S5 Fig). Nonetheless, the transition was not perfectly sharp, and the propagation speed of the sequence could be accurately predicted from the mean-field equations (S6 Fig). Examination of the activity distribution of the mean-driven sequence-generating network also suggests that the network "hops" between metastable states (S6 Fig), limiting the effective resolution of the network (i.e. the network can effectively be in one of $D$ states; S6 Fig). Thus, asymmetrically increasing mean weights between adjacent populations in an originally multistable network can produce rapid, stereotyped sequences accompanied by regular spiking. This possibly reflects a dynamical regime observed in neural circuits underlying song production in songbirds, where spike trains are thought to be highly regular and stereotyped [88]; however, it does not reproduce the irregular spiking of mammalian cortex.

**Fluctuation-driven sequence generation.** Sequences produced in the fluctuation-driven regime behaved quite differently. Indeed, unidirectionally increasing the weight variances $(\sigma_J^2)_2$ between adjacent populations (Fig 6a) of a fluctuation-driven multistable network also led to the production of sequences. As with the multistable case, our theory predicted a much smoother transition to sequence generation than in the mean-driven network (Fig 6b). This was corroborated by the full simulations, in which the propagation of the sequence could be easily controlled by setting $(\sigma_J^2)_2$, and which additionally suggested consistently irregular

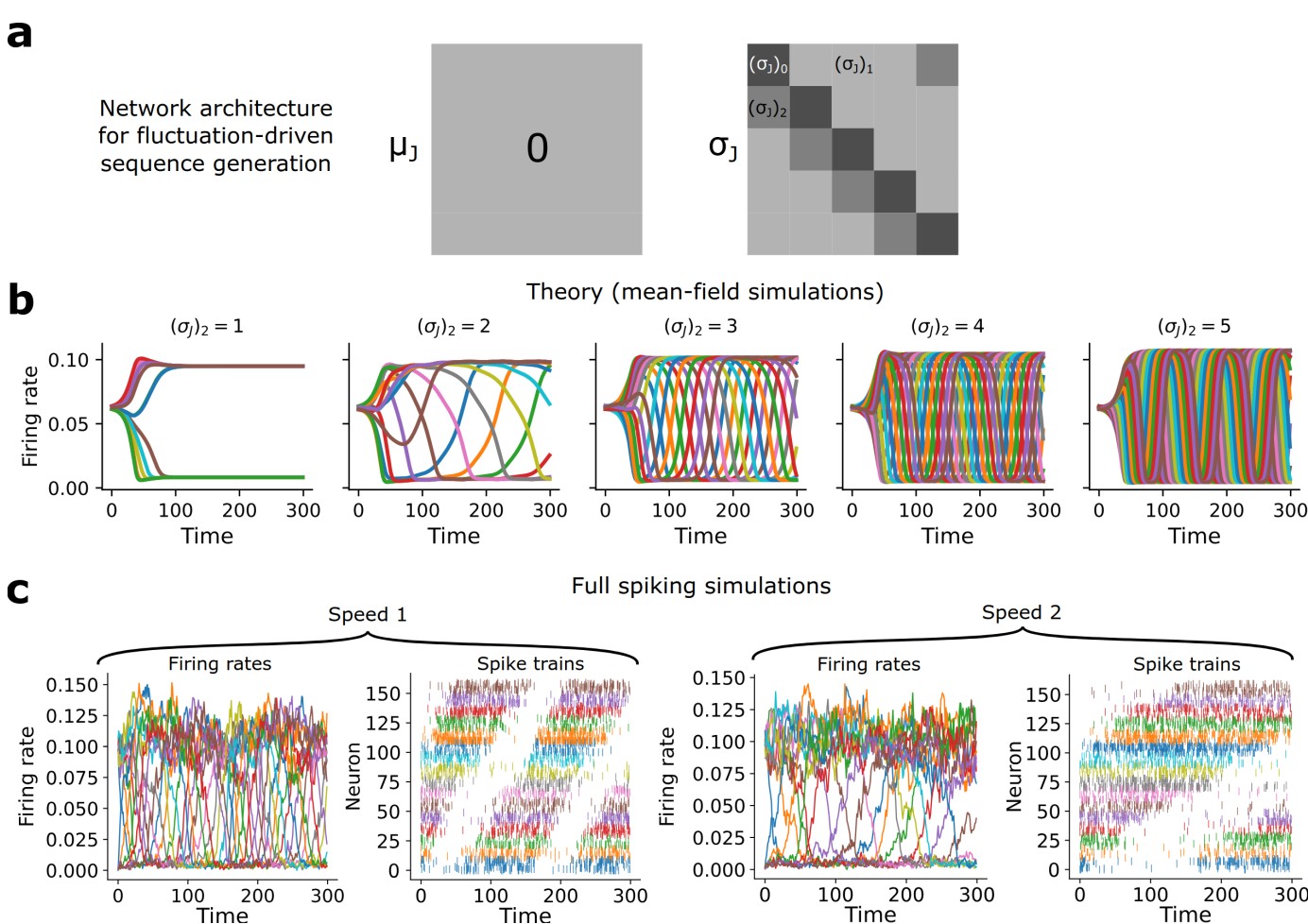

**Fig 6. Fluctuation-driven sequence generation.** (**a**) Schematic of network weights for fluctuation-driven sequence generation. (**b**) $N \to \infty$ mean-field simulations of firing rate dynamics in fluctuation-driven sequence generation, for 5 different values of $(\sigma_J)_2$, with $(\sigma_J)_0 = 10$, and $D = 16$. (**c**) Population firing rates and spike trains from an example full network simulation producing sequences at two different speeds (different $(\sigma_J)_2$, with $N = 2000, D = 16$).

spiking patterns throughout the sequence evolution (Fig 6c). The theory accurately predicted the speed of propagation in full simulations of the fluctuation-driven network (Fig 7a). Curiously, unlike the mean-driven case, the theory suggested that the activity pattern of the network should not hop between metastable states, but instead should "flow" more continuously around the ring (Fig 7b). This was corroborated in the full simulation, suggesting that the resolution of information stored in the sequence location may be higher than in the mean-driven network (Fig 7c). Finally, estimating the ISI $CV_2$ distribution from spike trains in the full simulation revealed high, Poisson-like irregularity (Fig 7d), suggesting that this network does not collapse to an unrealistic regular spiking regime when it produces sequences. As in the fluctuation-driven multistable case, irregular spiking in the fluctuation-driven sequence network emerges through chaos (S7 Fig). Thus, fluctuation-driven dynamics mediated by weight variances both produce and may advantage sequence generation, while also yielding consistently brain-like irregular spiking. Similar to the multistable network, this suggests that stable neural activity sequences can be produced without increased positive excitatory connections

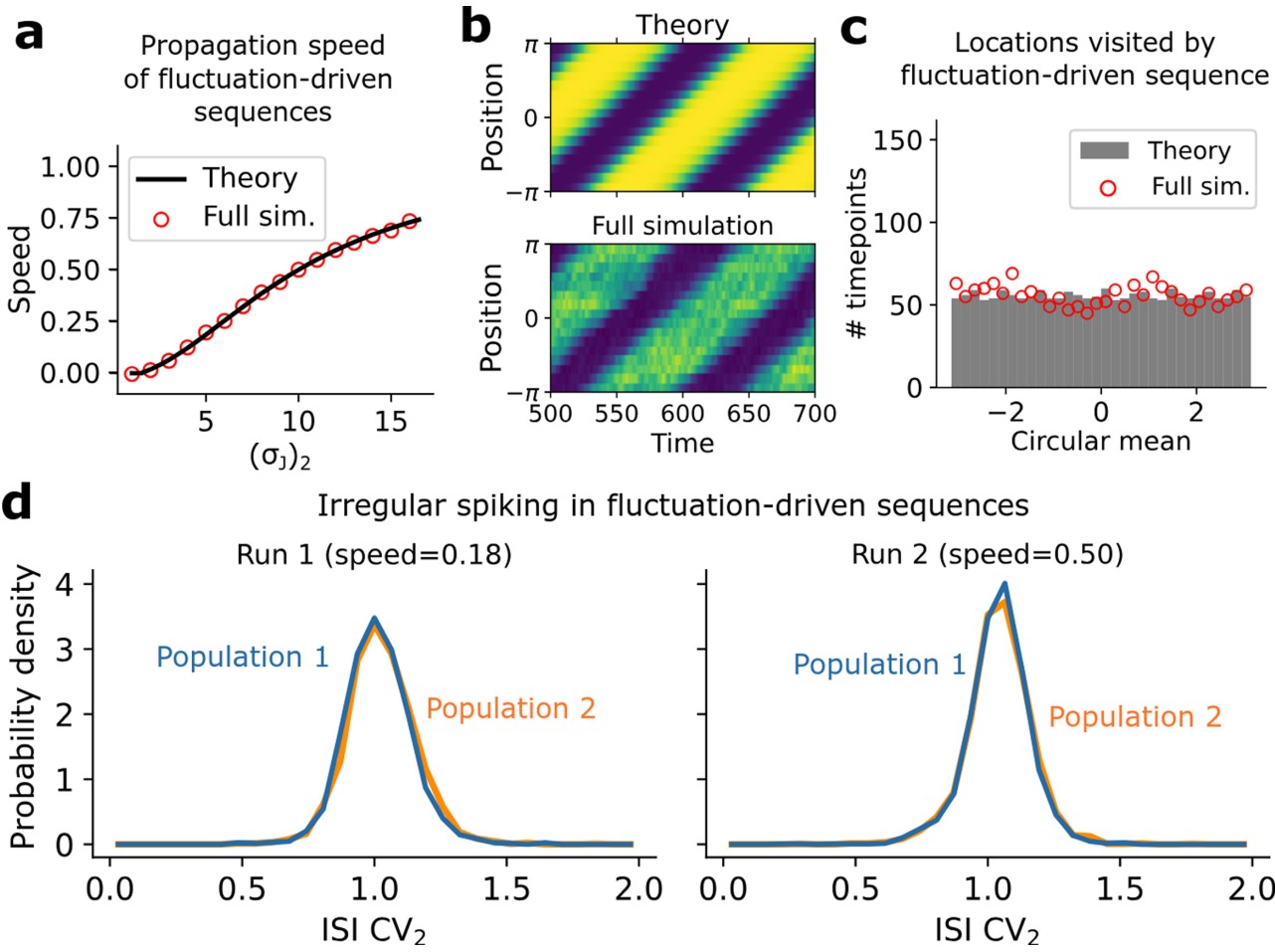

**Fig 7. Properties of fluctuation-driven sequence-generating network.** (**a**) Theoretical (mean-field; $N \to \infty$) predictions and full simulations ($N = 2000$, $(\sigma_J)_0 = 10$, $(\sigma_J)_1 = 1$) of sequence propagation speed in the fluctuation-driven network. (**b**) Heatmaps of the evolution of macroscopic network activity in the fluctuation-driven sequence-generating network, with $(\sigma_J)_2 = 4.2$, via either the $N \to \infty$ theory or the equivalent full simulation ($N = 2500$). (**c**) Histogram of circular means of macroscopic activity distribution in the fluctuation-driven sequence-generating network in (b). (**d**) ISI $CV_2$ distributions for neurons in two different populations in the fluctuation-driven sequence-generating network for two different propagation speeds.

between neurons—increased variances in connectivity can suffice instead, and are consistent with sequences mediated by irregular spiking.

## Heterogeneous fluctuation-driven firing rate dynamics

Finally, we asked whether the network could produce more complex firing rate dynamics $\mathbf{r}^t$ than multistability and sequences alone, yet still simultaneous with irregular spiking. Complex, time-varying firing rates are thought to provide a reservoir of temporal patterns that can be composed into flexible computations [2,28,42,89,90]. Typically, however, such dynamics are studied in "rate network" models of neural populations that do not explicitly account for spikes; most commonly, a $D$-dimensional vector of population firing rates, $\mathbf{r} = \phi(\mathbf{h})$, evolves according to $\tau d\mathbf{h}/dt = -\mathbf{h} + W\mathbf{r} + \mathbf{u}$, where $\mathbf{h}$ are private variables, $\mathbf{u}$ are inputs, $\tau$ is a time constant, $W$ is a $D \times D$ matrix of interaction weights between populations, and $\phi$ is an element-wise nonlinearity such as a tanh or sigmoid [1,2,5,33,90,91]. In contrast, our equations for $\mathbf{r}^t$

(Eq 5–10), which also describe the time-varying firing rates of $D$ populations, operate fundamentally differently—besides being in discrete time, Eq 5–10 are parameterized by two $D \times D$ matrices, $\mu_J$ and $\sigma_J^2$, and are mediated by a *non*-element-wise nonlinearity (Eq 5). Thus, it is unclear how similarly they behave to the more common rate network equations. Note also that while the equations for $\mathbf{r}^t$ are derived (as the mean-field description of the full network dynamics) under the assumption of irregular spiking in $\{\mathbf{y}_i^t\}$, irregular single-neuron activity does not imply irregular or complex dynamics in $\mathbf{r}^t$; for instance, in the unstructured network the spikes are irregular but the population firing rates are constant (Fig 2 and 3). Here, we investigate whether irregular activity in $\{\mathbf{y}_i^t\}$ can co-exist with complex, time-varying dynamics of the population firing rates $\mathbf{r}^t$.

To address this problem, we simulated the mean-field equations for $\mathbf{r}^t$ (Eq 5, 9, 10; $N \to \infty$ limit), fixing $\mu_J = 0$ and only allowing ourselves to vary the matrix parameter $\sigma_J^2$. This procedure keeps the network in the fluctuation-driven regime (since all mean inputs are 0; Eq. 6, 9), which is in turn expected to produce irregular spiking in the corresponding full simulations. A challenge of this approach, however, is that whereas the matrix $W$ in classic rate networks can usually contain both positive and negative values [1,2,5,90,91], all elements of $\sigma_J^2$ are non-negative by definition. (Note that in our full model, the weights $J_{ij}^{dd'}$ between individual neurons can be both positive and negative [Eq 4]—the non-negative matrix $\sigma_J^2$ specifies the variances of $J_{ij}^{dd'}$, in both the mean-field equations for $\mathbf{r}^t$ and the full model, and is the only parameter we vary in the present simulations of $\mathbf{r}^t$.) Given this constraint, it is not obvious that Eq 5–10 could produce activity with the same level of complexity as can arise in classic rate networks. We hypothesized, however, that if $\sigma_J^2$ were a sparse random matrix, the normalization of the network activity through Eq 5 would play a similar role as the inhibitory feedback in classic rate networks.

As with the sequence-generating simulations, to test our network's ability to generate complex firing rate dynamics we started with a multistable network, then destabilized it. Specifically, we first set all diagonal entries of $\sigma_J^2$ to 1 and the rest to 0 (which yields multistability), then outside the diagonal introduced $K_\sigma$ nonzero elements in each row, each with a value of $(\sigma_J^2)_r$ (Fig 8a). We predicted that when the density of nonzero entries in the weight variance matrix $\sigma_J^2$ was sufficiently high, but still sparse relative to the total number of populations $D$ (i.e. $0 \ll K_\sigma \ll D$), the network would be forced into a regime in which strong fluctuation-driven feedback competes with this normalization to produce chaotic-like dynamics, in loose analogy to how tanh or sigmoidal "squashing" nonlinearities transform unstable linear dynamics into chaos in classic rate networks [1].

Simulating the mean-field equations for $\mathbf{r}^t$ for a variety of $D$, $K_\sigma$, and $(\sigma_J^2)_r$ revealed a rich repertoire of firing rate dynamics. For instance, for $D = 100$ and small $K_\sigma$ and $(\sigma_J^2)_r$, the network remained multistable, with multiple firing rate levels corresponding to active states (Fig 8b, left). For a larger $D = 200$, and stronger recurrent weight variances ($K_\sigma = 5$, $(\sigma_J)_r = 0.3$), certain initial conditions additionally produced extended firing rate dynamics in which different populations became active in a haphazard fashion, although fixed points could still arise (Fig 8b, middle). When $K_\sigma$ was even larger, the firing rate dynamics exhibited consistently strong spatiotemporal heterogeneity (Fig 8b, right). Thus, similar to classic rate networks [1,2], strong recurrent feedback can also produce highly variable, heterogeneous firing rate dynamics—the mechanism and corresponding dynamical system are different in our model, however, since all firing rate dynamics follow from dynamics of the input variances $\{\sigma_d^2\}$, with mean inputs $\{\mu_d\}$ always zero (Eq 6–10).

Finally, we ran the full spiking network simulations (Eq 1–4) corresponding to an example parameter regime in which the equations for $\mathbf{r}^t$ produced temporally complex firing rate

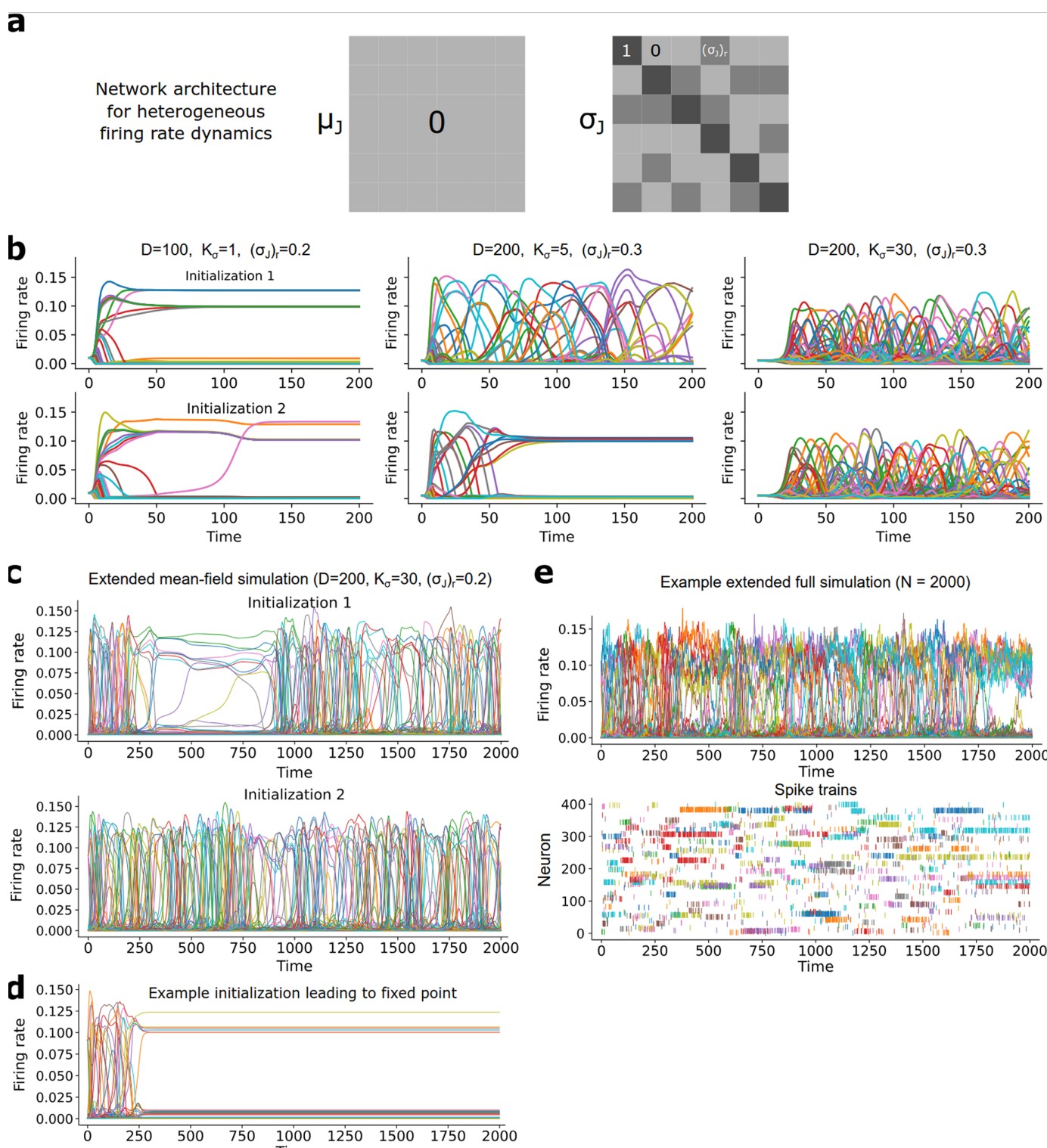

**Fig 8. Heterogeneous fluctuation-driven firing rate dynamics. (a)** Schematic of network configuration. **(b)**. Simulations of mean-field dynamics for $\mathbf{r}^t$ for different parameter sets corresponding to the architecture shown in (a). **(c)** Extended mean-field simulation ($N \to \infty$) of fluctuation-driven firing rate dynamics produced by an example network starting from 2 different initial conditions that lead to complex rate dynamics. **(d)** Example initial condition from the same network in (c) leading to a fixed point in the mean-field dynamics. **(e)** Full spiking network simulation showing firing rate dynamics and spike train dynamics with macroscopic parameters equivalent to the simulation in (c–d). Spike trains are shown for two example neurons per population.

dynamics. We first identified an example set of parameters (using the mean-field dynamics [Eq 5, 9, 10]) in which random initial conditions frequently produced extended complex firing rate dynamics in the $N \to \infty$ case (Fig 8c). This particular network instantiation also sometimes exhibited fixed points in the dynamics (Fig 8d), suggesting the possible coexistence of chaos (or at least complex transients) with fixed points in a single network, although further analyses will be required to characterize this in detail. We then ran the corresponding full spiking network simulation, which produced similar dynamics as the mean-field predictions (Fig 8e), exhibiting slow, heterogeneous changes in firing rates that continued over many timesteps. Both the mean-field and full spiking network simulations yielded firing rates with wide autocovariance functions (Fig 9a and 9b), extending far beyond the single-neuron memory timescale ($\Delta t = 1$) in the microscopic update rule (Eq 1, Eq 3). Note that in the full spiking simulations some of the dynamics may arise from finite-size effects, similar to how finite numbers of neurons in balanced clustered networks allow spontaneous fluctuations to drive transitions between metastable states [26–28]. However, because the slow timescales persist in our mean-field simulations (computed in the $N \to \infty$ limit), the slowness and complexity of the firing rate dynamics are likely not exclusively a finite-size effect in $N$. Intuitively, the slowness is inherited from the multistable network, with the additional couplings between the mean-field populations via the nonzero entries of $\sigma_J^2$ destabilizing the multistability. Further, the existence of slow firing rate dynamics given different network parameters (Fig 8b) suggests that fine-tuning is not required.

To assess whether the model also produced irregular spikes we examined the temporal patterning of the spikes generated in the full network simulations. As we constructed our mean-field simulations for $\mathbf{r}^t$ to operate in the fluctuation-driven regime, the spikes generated

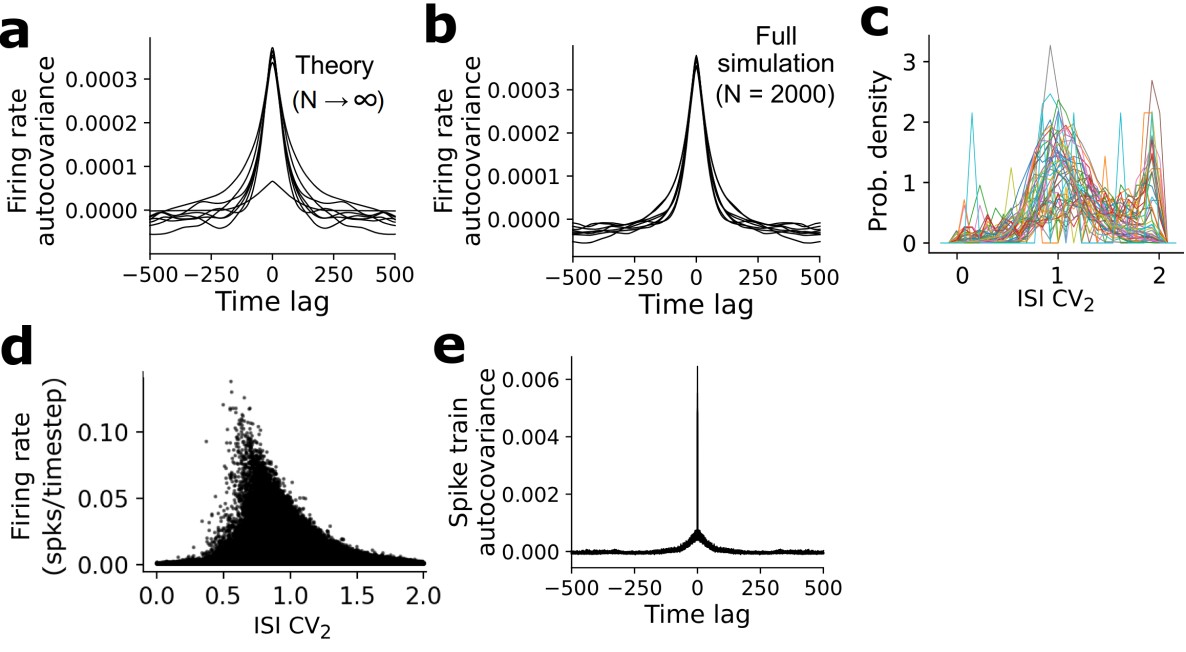

**Fig 9. Properties of network producing heterogeneous fluctuation-driven firing rate dynamics.** (**a**) Sample firing rate autocovariance function from the infinite mean-field simulations in Fig 8c, averaged across all co-tuned populations (for 8 example initializations of the same network). (**b**) As in (**a**) but computed from the full spiking network simulation ($N = 2000$) (**c**). ISI $CV_2$ distributions for several example co-tuned populations from Fig 8e. (**d**). Firing rate vs ISI $CV_2$ for all neurons in the network. (**e**) Spike-train autocovariance function, averaged across all neurons (each of the 8 traces shows a different initialization, although the traces are very similar).

in the corresponding full network simulation of $\{\mathbf{y}_i^t\}$ were highly irregular (Fig 9c). Although some neurons substantially deviated from the Poissonian signature of ISI $CV_2 \approx 1$, these neurons spiked rarely in the simulation, hence their spike trains were probably undersampled (Fig 9d). The autocovariances of the spike trains were a combination of a delta-function-like peak at a zero time-lag, indicative of Poisson-like firing, together with the wider base reflecting the slow evolution of the firing rates (Fig 9e). Thus, the fluctuation-driven regime of our network can produce brain-like irregular spike trains together with complex, heterogeneous, and slowly varying firing rate dynamics; which our theory suggests should persist as $N \to \infty$. A detailed analysis of the dependence of this regime on $D$ and further features of $\sigma_J^2$ and $\mu_J$, as well as the possibility of chaotic firing rate dynamics, will be an important direction for future work.

## Discussion

We have shown how recurrent networks of winner-take-all (WTA) units can produce brain-like irregular spiking capable of powering flexible nonlinear collective dynamics. In contrast to the more typical spike-threshold nonlinearity, the WTA nonlinearity causes the network to avoid the emergent linear dynamics of classic balanced networks [11,17,18]. In turn, programming the network by adjusting the weight variances alone causes firing rate dynamics to follow the dynamics in the input variances, rather than the means, which in turn yields consistently irregular spiking. Our model thus reveals the dynamical consequences of WTA as a repeated functional motif in a large network and represents an alternative model class (compared to excitatory-inhibitory balanced networks [1,11,17,19,26,27,40,41,44,92]) for studying biological neural activity, which unifies Poisson-like temporal irregularity with flexible nonlinear collective dynamics.

Our mean-field equations for the firing rates $\mathbf{r}^t$ describe deterministic, nonlinear dynamics even in the $N \to \infty$ limit (Eq 5, 9–10). Thus, the network's nonlinear behavior holds when neurons receive an infinite number of synaptic inputs. This contrasts with finite-size theories, e.g. the theory of "loose balance," which reproduces empirical nonlinear phenomena as a consequence of neurons receiving finite synaptic inputs [11,39,93]. A related model [24] showed how single-neuron response-onset and saturation nonlinearities could partially propagate to the network level if recurrent interactions were weak, but this network also relied on finite synaptic inputs or weak couplings. Nonlinearity can also emerge in "semi-balanced" networks, in which populations can be temporarily silenced by excess inhibition, leading them to behave as rectified linear units [23]; our network, however, does not strictly require any population to be silent to produce nonlinear behavior (Fig 4b). Clustered balanced networks can produce multi- or metastable dynamics [26,27], but active clusters typically generate more regular spikes, whereas the active populations in our multistable network produce highly irregular spikes (ISI CV $\sim$ 1). Certain short-term plasticity processes can yield bistable collective activity, with irregular spiking persisting in active states—as in our model, irregular spiking follows from state-dependent input variances [40]; however, the role such plasticity processes play in cortex is not well established, and it is moreover unclear how to generalize [40] to produce more complex dynamics such as sequences (Fig 6). An alternative model showed how correlated noise could drive sequences of state transitions matching empirical temporal variability [87], but this network did not include spikes—it will be interesting to explore whether correlated noise can facilitate dynamics in our network as well. In general, it will be interesting to compare these and other balanced network models [11,20,21,23,24,40, 41,43,94,95] with our network in terms of the collective dynamics they can produce and their relationship to empirical neural dynamics.

## Biological basis

The key feature of our network is the Softmax/WTA nonlinearity operating within each unit, in which neurons with different tunings compete to activate [46]. This competition was inspired by biology [46] and could emerge through a variety of mechanisms, such as lateral inhibition, recurrent amplification, or synaptic depression [46,48,49,55,96]. Another possibility is via axo-axonic inhibition, in which one excitatory cell projects directly onto an inhibitory terminal onto a second excitatory cell, providing a means for inhibition without requiring the inhibitory cell to spike [97]. In principle, competition within a unit could also occur between different activity patterns rather than different neurons, with these patterns competing in a WTA-like fashion; or similarly between attractor-like states in small cortical patches [98]. An important consideration for more detailed network models is the timescales over which the key interactions unfold. In particular, we have modeled the integration of presynaptic inputs and the WTA operation as occurring within one timestep, roughly corresponding to one membrane time constant (10-20 ms [62]). Indeed, neurons integrate their presynaptic inputs over this timescale [62], and it has been shown that WTA-like operations can occur at or faster than this timescale in networks of integrate-and-fire neurons [48], suggesting the microscopic timescales in our model are consistent with biology. Formally extending the network to operate with continuous-time integrate-and-fire neurons that compete within small groups while also receiving fluctuating inputs will be important future work.

A second important direction is to determine how the network would be implemented with neurons that are strictly excitatory or inhibitory. In our model any neuron can send both positive and negative projections to other neurons in the network. While this violates Dale's law (the restriction that each neuron should be strictly excitatory or inhibitory [99]), such an approach is often used as a first-pass approximation when studying collective neural dynamics [1,5,70,100,101]. The key assumption in our theoretical treatment is that inputs to any neuron at any timestep are i.i.d. Gaussian—therefore we expect our results to also hold when each neuron is strictly excitatory or inhibitory, so long as connectivity is sparse enough that inputs across different cells are nearly uncorrelated [19] and the network is large enough that the central limit theorem holds. Biologically, our network model is also consistent with multiple roles for inhibition, mediating (1) interactions between units as well as (2) the competition within a unit, and which might be supported biologically by different inhibitory cell types [102–104].

## Training and fitting to data

The network we have presented can produce flexible macroscopic dynamics, depending on the $D \times D$ matrices $\mu_J$ and $\sigma_J^2$. Thus, beyond the specific regimes we have studied here, the network's many parameters ($2D^2$ in total) can also be generically adjusted to reproduce specific target dynamics that perform computations, possibly via e.g. gradient methods like back-propagation through time [105] or reservoir-computing methods [2,89,106]. The network's flexibility suggests it may in turn share key features with traditional non-spiking recurrent neural networks [1,2], notably the ability to transform input time-series into rich, high-dimensional nonlinear representations to facilitate flexible learning. The tendency of the fluctuation-driven regime to avoid saturation, however, may endow the network with additional abilities that challenge traditional networks, such as the handling of long-term dependencies [107]. A detailed comparison of the computational abilities of our network, traditional RNNs, as well as other modern recurrent neural networks [108,109] specializing in handling long timescales, will be an exciting future research direction.

In principle one can also fit our network to data using maximum likelihood methods. Recent experimental technologies now allow for large-scale recordings of population spike trains over extended periods [110–112]. Modeling the spiking produced by our network as i.i.d. Bernoulli conditioned on firing rates (a slight simplification since firing rates are heterogeneous within a population and neurons in individual units negatively correlate), we can write the conditional probability of a binarized population spike train recording $\mathbf{s} \in \{0,1\}^{T\times n}$, of length $T$ from $n$ neurons, as

$$P(\mathbf{s}|\mathbf{r}^{1:T}) = \prod_{k,t} \text{Bern}\left[s_k^t|r_{\tilde{d}_k}^t\right] \tag{12}$$

where $\mathbf{r}^{1:T} \in \mathbb{R}^{T\times D}$ are the $D$ population firing rates from time 1 till $T$, $t$ and $k$ index time and neurons, and $\tilde{d}_k$ indicates the population to which neuron $k$ belongs. The probability of a population recording $\mathbf{s}$, i.e. the marginal likelihood of the network parameters, is then

$$P(\mathbf{s}|\mu_J, \sigma_J^2) = \int \prod_{k,t} \text{Bern}\left[s_k^t|r_{\tilde{d}_k}^t\right] P(\mathbf{r}^{1:T}|\mu_J, \sigma_J^2) \prod_t d\mathbf{r}^t. \tag{13}$$

Here $\mathbf{r}^{1:T}$ plays the role of a latent variable, much like the hidden state of a Hidden Markov Model (HMM) or Kalman filter [113–115]—in fact, the dynamics of the firing rates (Eq 5–10) given the network parameters are Markovian, since they depend only on the previous timestep. The model could therefore potentially be fit with a variation of Expectation Maximization [116], which alternates between inferring the latent sequence (here $\mathbf{r}^{1:T}$) and learning the parameters (here $\mu_J, \sigma_J^2$), and which is commonly applied to HMMs. Note that here one also requires a procedure for learning or marginalizing over the hyperparameters $D$, $N$, and $\{\tilde{d}_k\}$.

Thus, our proposed network represents a new statistical model for analyzing population spike train data. A key advantage over most latent variable models, which usually treat spikes as noisy readouts of abstract latent dynamics [9,10,117–119] (although see [120]), is that the fit model can be directly translated into a self-contained spiking network. Therefore any computational dynamics inferred by fitting the network to data are automatically compatible with a mechanistic model operating at the level of individual spiking neurons (grouped into WTA units), and which as we have shown is well-suited for implementing complex dynamics through irregular spike trains.

## Limitations

Our model as instantiated has important limitations that should be addressed in future work. First, although our fluctuation-driven results (Fig 4–9) suggest that these dynamics, most notably the slow timescales, do not require fine-tuning but rather change gracefully as the parameters are varied, the model may still be sensitive to other features of its construction. It will be particularly useful to examine robustness against perturbations of the competitive WTA nonlinearity, e.g. if competition is not symmetric among neurons in a unit but spreads across units, possibly asymmetrically; if not every unit contains every tuning; or if neurons have continuous and/or overlapping tunings rather than discrete tunings. Second, the firing rates that drive nonlinear computation are averages over neurons within a population—this stands in contrast to approaches for constructing spiking networks where single rate units are converted to single spiking neurons [42,121–123], which can in principle make more efficient

use of individual neurons. Thus, an interesting future direction will be to examine how heterogeneity of activity within each population may support additional computational dynamics [124], such that the effective dimensionality of the system computation extends beyond the number of populations and toward the number of neurons.

## Predictions

Despite its limitations and idealized nature our model makes testable predictions. First, we predict that neurons with correlated slow changes in firing rates (i.e. with similar "tuning") should have higher *variance* functional couplings at millisecond timescales, but not necessarily higher mean couplings, as in our model the former are the source of slow timescales. This could be tested by fitting a network-generalized-linear-model [125] to population spike trains and comparing the inferred coupling filters between neuron pairs with the strength of correlation of their firing rates. A second prediction is that in large neural recordings computational dynamics will unfold primarily without affecting the mean firing rate across the full population of recorded neurons (which was fixed at 1/D in our model), which has been observed in monkey prefrontal cortex [38] and in mouse and monkey visual cortex [63]. Third, we predict that nonlinear firing rate dynamics underlying neural computations are significantly fluctuation-driven, rather than only mean-driven. While this prediction could be tested via *in vivo* intracellular recordings, one could more easily test it by fitting the network model to large-scale population spike trains [110–112], where we predict that the goodness of fit will rely heavily on structure in the weight variances. Our final core prediction is that empirical neural dynamics, in particular in awake cortical recordings, will be better described by rate equations closer to our mean-field equations (Eqs 5–10) than to more common model classes, e.g. of the form $\tau d\mathbf{h}/dt = -\mathbf{h} + W\phi(\mathbf{h}) + \mathbf{u}$, where $\mathbf{r} = \phi(\mathbf{h})$ are the firing rates and $\phi$ is an element-wise monotonic nonlinearity [1,2,5,33,90,91]. This could be tested by fitting both models to data, treating the firing rates as latent variables that produce observed spikes through a Poisson process, and comparing the fits.

## Methods

### Mean-field theory summary

Here we give a summary/intuition behind the derivation of the mean-field dynamics for the firing rates (Eq 5–10). Briefly, given $\mathbf{r}^{t-1}$, this means that approximately $n_{d'}^{t-1} = r_{d'}^{t-1}N$ neurons in population $d'$ just spiked at $t-1$. We suppose that these $n_{d'}^{t-1}$ neurons are uniformly sampled from that population (note that this assumption ignores heterogeneity of firing rates within the population). The total input to a postsynaptic neuron with tuning $d$ from population $d'$ is then approximately a sum of $n_{d'}^{t-1}$ i.i.d. Gaussian inputs with mean $\mu_J(d, d')D/N$ and variance $\sigma_J^2(d, d')D/N$, since we have now randomly sampled and summed $n_{d'}^{t-1}$ elements from the quenched weight matrix $J_{ij}^{dd'}$, which were constructed as i.i.d. (Eq 4), so the resulting sum is itself Gaussian. The total input to a postsynaptic neuron will then be the sum of $D$ such inputs from the $D$ populations, with possibly different means and variances, which is again Gaussian. Thus, at time $t$ each neuron $d$ within a unit receives a Gaussian input with a mean and variance determined by $\mathbf{r}^{t-1}, \mu_J$, and $\sigma_J$, and the probability of that neuron spiking is the probability that its input is larger than the input to every other neuron in the unit. This probability is equal to the firing rate $r_d^t$ of population $d$, which once computed for all $D$ populations gives us $\mathbf{r}^t$. A full derivation can be found in S1 Appendix.

## Simulations

For the most part, networks were simulated directly using Eqs 1–4. For very large networks, however, we approximate the network connectivity $J^{dd'} \in \mathbb{R}^{N \times N}$ where $d, d' \in 1, ..., D$ by storing a single connectivity block $J_0 \in \mathbb{R}^{N \times N}$ in memory and generating the remaining blocks at run time via quenched random permutations of $J_0$ and systematic scaling/shifting for each $d, d'$. Details can be found in S2 Appendix.

## Estimation of eigenvalues

In Fig 4 we numerically estimated the eigenvalue $\lambda_{max}$ (of the Jacobian of the mean-field dynamics evaluated at the uniform firing rate distribution $\mathbf{r}_d = 1/D$), using a step size of $\epsilon = 10^{-12}$.

## Nonstationary coefficient of variation of interspike intervals

The nonstationary coefficient of variation of interspike intervals, or ISI $CV_2$, quantifies how similar interspike intervals are to those of an inhomogeneous Poisson process [14]. It is given by

$$\text{ISI CV}_2 = \sum_l \frac{2|\text{ISI}_{l+1} - \text{ISI}_l|}{\text{ISI}_{l+1} + \text{ISI}_l} \tag{14}$$

where $\text{ISI}_l$ is the $l$-th interspike interval in a spike train. A value of ISI $CV_2 = 1$ is a signature of Poissonian spiking.

## Numerical estimation of transition to chaos

We estimated the transition to chaos in Fig 1d by (1) running a simulation of the full network for 10 timesteps, (2) creating a copy of the network and its activity state at $t = 10$, (3) picking a single WTA unit in the copied network and swapping the activity levels of the most and least active neuron in that unit (a microscopic perturbation), (4) running both networks forward for 20 more timesteps, (5) computing the Euclidean distance between the complete states of the two networks as a function of time, (6) labeling the network chaotic if the distance increased following the perturbation and non-chaotic if it decreased.

## Supporting information

**S1 Fig. Sensitivity of the uniform balanced state to microscopic perturbations. a**. Root-mean-square distance between the microscopic network state ($\{\mathbf{y}_d^t\}$) vs time since a single spike-swap perturbation is introduced. Parameters used were: $D = 16$, $N = 2000$; $\mu_J(d, d') = 0 \quad \forall d, d'$; $\sigma_J^2(d, d') = 1 \quad \forall d, d'$. **b-c**. Firing rates corresponding to the reference run (**b**) vs the perturbed run (**c**, in which the spike swap is introduced at t=0).
(PDF)

**S2 Fig. Mean-driven multistability. a**. Schematic of network weights for mean-driven multistability. **b**. Mean-field simulations ($N \to \infty$) of firing rate dynamics for 6 different $(\mu_J)_0$. **c**. Example firing rate dynamics and spike trains from a full simulation of the mean-driven multistable network ($(\mu_J)_0 = 3$, $N = 3000$, $D = 16$).
(PDF)

**S3 Fig. Properties of mean-driven multistable network. a**. Maximum eigenvalue $\lambda_{max}$ of the Jacobian of the $N \to \infty$ mean-field dynamics evaluated at the uniform distribution $\mathbf{1}/D$ as a

function of $D$ and $(\mu_J)_0$. **b**. Number of active populations (top) or equivalently information content (bottom) for the mean-driven network ($(\mu_J)_0 = 1.5$). Solid lines are theory; circles are full simulation results ($N = 3000$).
(PDF)

**S4 Fig. Sensitivity to spike perturbations in the fluctuation-driven multistable network.**
**a**. Root-mean-square distance between the microscopic network state ($\{\mathbf{y}_d^t\}$) vs time since a single spike-swap perturbation is introduced, in the fluctuation-driven multistable network. Trace and error bars show mean and standard deviation, respectively, over 10 trials. Parameters used were $N = 2000$, $D = 16$, $(\sigma_J)_0 = 1$, $(\sigma_J)_1 = 0.05$. **b**. Evolution of firing rates for a reference simulation and a perturbed simulation (created by a single spike swap, relative to the reference network, at time 0). **c**. As in b, but for another trial.
(PDF)

**S5 Fig. Mean-driven sequence generation. a**. Schematic of network weights for mean-driven sequence generation. **b**. $N \to \infty$ mean-field simulations of firing rate dynamics in mean-driven sequence generation, for 5 different values of $(\mu_J)_2$, with $(\mu_J)_0 = 3$, and $D = 16$. **c**. Population firing rates and spike trains from an example full network simulation with $(\mu_J)_0 = (\mu_J)_2 = 3, (\sigma_J)_0 = (\sigma_J)_1 = 1$ and $N = 2000, D = 16$.
(PDF)

**S6 Fig. Properties of mean-driven sequence-generating network. a**. Theoretical predictions and full simulations ($N = 2000, (\mu_J)_0 = 3$) of sequence propagation speed in the mean-driven network. **b**. Heatmaps of the evolution of macroscopic network activity (firing rates) in the mean-driven sequence-generating network, via either the $N \to \infty$ theory or a full simulation ($(\mu_J)_0 = 3, (\mu_J)_2 = 2.3, N = 2500$). **c**. Histogram of circular means of macroscopic activity distribution in the mean-driven sequence-generating network.
(PDF)

**S7 Fig. Sensitivity to spike perturbations in the fluctuation-driven sequence network a**. Root-mean-square distance between the microscopic network state ($\{\mathbf{y}_d^t\}$) vs time since a single spike-swap perturbation is introduced, in the fluctuation-driven sequence-generating network. Trace and error bars show mean and standard deviation, respectively, over 10 trials. Parameters used were $N = 2000, D = 16, (\sigma_J)_0 = 10, (\sigma_J)_1 = 1, (\sigma_J)_2 = 10$. **b**. Evolution of firing rates for a reference simulation and a perturbed simulation (created by a single spike swap, relative to the reference network, at time 0). **c**. As in b, but for another trial.
(PDF)

**S1 Appendix. Full derivation of the mean-field dynamics for the firing rates.**
(PDF)

**S2 Appendix. Approximate simulations for large networks.**
(PDF)

## Acknowledgments

I would like to thank Agostina Palmigiano, Tatiana Engel, Rainer Engelken, Lee Susman, Luca di Carlo, Kamesh Krishnamurthy, Yiqin Gao, Tankut Can, Daniel Weilandt, Nicolas Lenner, and Manuel Schottdorf for insightful conversations about this project, as well as other members of the Engel lab. Computational resources, office space, and conference funding to present earlier versions of this work were provided by the National Science Foundation through the Center for the Physics of Biological Function (PHY-1734030). This work was conceived during the 2019 IBRO-Simons Computational Neuroscience Imbizo.

## Author contributions

**Conceptualization:** Rich Pang.

**Formal analysis:** Rich Pang.

**Investigation:** Rich Pang.

**Methodology:** Rich Pang.

**Project administration:** Rich Pang.

**Software:** Rich Pang.

**Validation:** Rich Pang.

**Visualization:** Rich Pang.

**Writing – original draft:** Rich Pang.

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
