## [Decision Letter · Decision Letter 0]

PCOMPBIOL-D-24-02146

Balanced state of networks of winner-take-all units

PLOS Computational Biology

Dear Dr. Pang,

Thank you for submitting your manuscript to PLOS Computational Biology. After careful consideration, we feel that it has merit but does not fully meet PLOS Computational Biology's publication criteria as it currently stands. Therefore, we invite you to submit a revised version of the manuscript that addresses the points raised during the review process.

Please submit your revised manuscript within 30 days Mar 29 2025 11:59PM. If you will need more time than this to complete your revisions, please reply to this message or contact the journal office at ploscompbiol@plos.org. Please include the following items when submitting your revised manuscript:

We look forward to receiving your revised manuscript.

Kind regards,

Jonathan Rubin

Academic Editor

PLOS Computational Biology

Daniele Marinazzo

Section Editor

PLOS Computational Biology

**Journal Requirements:**

3) We notice that your supplementary Figures, and information are included in the manuscript file. Please remove them and upload them with the file type 'Supporting Information'. Please ensure that each Supporting Information file has a legend listed in the manuscript after the references list.

**Reviewers' comments:**

Reviewer's Responses to Questions

Reviewer #1: This manuscript presents a novel model of interacting populations of winner take all networks. The authors present a mathematical analysis of the networks in the high gain limit and show that the networks can produce asynchronous irregular spiking activity, sequence generation, and other common features of neural activity in the brain. The model is interesting and, as far as I am aware, novel. Given the novelty of this approach, the authors do a good job of laying the groundwork for future work on developing these models, which could hold promise for better understanding neural circuit dynamics.

I only have a handful of minor comments:

1) The title of the manuscript and the body of the manuscript (esp the Introduction) mention "balance" but it seems that the authors are using this term at least in part to mean asynchronous-irregular activity. In traditional network models (e.g., integrate-and-fire or binary network models), excitatory-inhibitory balance often coincides with asynchronous-irregular activity and EI balance was historically studied as a mechanism to generate asynchronous-irregular activity. However, they do not always coincide and they are fundamentally different phenomena. Network models can exhibit EI balance without asynchronous-irregular activity or asynchronous-irregular activity without EI balance. The authors mention some notion of "balance" for example when they write (line 159) "the balance of inputs from different co-tuned populations" but this seems fundamentally different from excitatory-inhibitory balance, which is what the term "balance" normally refers to in the field. If there is a direct connection between these two notions of balance, the authors should clarify it. If not, the authors should avoid conflating the two (e.g., (line 32) "we present a generalization of the classic balanced state"; in what sense is their model a generalization?). In general, I think that the relationships between asynchronous-irregular activity, excitatory-inhibitory balance, and whatever notion of balance their models exhibit should be made more clear. And the authors should avoid statements that conflate these phenomena. I realize that this is a somewhat open-ended suggestion and I will not nitpick exactly how the authors choose to address it. This is more of a suggestion.

2) Line 100: It might be more accurate to change the two appearances of the work "any" to "all"?

3) Lines 127-133: It might be worth mentioning or at least citing the work of Monteforte and Wolf here:

Monteforte, Michael, and Fred Wolf. "Dynamic flux tubes form reservoirs of stability in neuronal circuits." Physical Review X 2.4 (2012): 041007.

Monteforte, Michael, and Fred Wolf. "Dynamical entropy production in spiking neuron networks in the balanced state." Physical review letters 105.26 (2010): 268104.

Engelken, Rainer, Michael Monteforte, and Fred Wolf. "Sparse chaos in cortical circuits." arXiv preprint arXiv:2412.21188 (2024).

They studied similar notions of "effective" chaos. The idea, I think, is that the network is not strictly speaking chaotic for fixed N, but becomes chaotic as N goes to infinity (though it's effectively chaotic in practice for relatively small N). I imagine something similar is happening in these models where a finite state space prevents true chaos, but as the state space grows, the dynamics approximate chaos.

4) Lines 227-229: When discussing the linearity/nonlinearity of rates in the balanced state, it might be worth citing Zhu, Baker, and Rosenbaum:

Baker, Cody, Vicky Zhu, and Robert Rosenbaum. "Nonlinear stimulus representations in neural circuits with approximate excitatory-inhibitory balance." PLoS computational biology 16.9 (2020): e1008192.

5) Lines 271-275: When discussing multistability in balanced networks, it could be worth citing the same paper by Zhu et al, in which multiple stable fixed points can arise in a generalization of balanced networks.

6) For many equations in the manuscript (e.g., Eq 2), the text following the equation is indented even though there is no paragraph break. The authors can avoid this by not skipping a line in their TeX code after the equation.

General discussion/comments that the authors are free to ignore:

1) General comment (pertaining in part to Lines 577-592): The authors discuss some predictions and interpretations of their results for neural circuits. It seems their model also predicts the presence of co-tuned populations of neurons. Of course, co-tuned neurons exist in neural circuits. But do these models make any additional predictions about the shared properties or dynamics of co-tuned populations beyond just similar preferred stimuli? As a related but separate question, how might this model be consistent with the presence of complex cells whose response properties are not described by a single preferred stimulus? These are open-ended questions/suggestions that the authors are free to ignore and save for future work.

2) General comment (pertaining to Lines 497-518): The authors discuss some interpretations of their model in terms of neural circuits (e.g., lateral inhibition). An additional interpretation is that each "neuron" in their models might represent something more abstract in a real neural circuits, e.g., a linear projection of neural activity in a population in some other direction (as in a distributed code under WTA dynamics). If the authors want, they could discuss this possibility, but this is a completely optional suggestion.

Reviewer #2: This is a very interesting paper.

The author proposes a model network that reconciles temporal irregular activity at the microscopic level with non-linear dynamics at the macroscopic level. Non-linear dynamics is clearly a sine qua non for computation. On the other hand, temporal irregularity is an experimental constraint that any suitable model/theory of cortical function must reproduce. The standard account of temporal irregularity is the theory of balanced networks that in its common implementation, however, does not produce non-linear dynamics at the macroscopic level (e.g., multi-stability). Essentially, this is because the synaptic input driving the activity of a neuron is a linear function of the pre-synaptic 'firing rates'. In the present contribution, this linearity is 'destroyed' by the winner-take-all (WTA) mechanism within a 'unit'; de facto, the activity of a given neuron depends not only on its own synaptic input but on the synaptic inputs to all the neurons in the same unit.

In the manuscript, the author illustrates the different dynamical regimes that such networks can exhibit and develops an accompanying mean-field theory that is shown to well capture some quantitative aspects of the relevant dynamics. The paper is well written. The results are novel to the best of my knowledge and, as I have already said, interesting. I have a few minor comments that mostly amount to the clarification of a few points.

I was confused by the section "Heterogeneous fluctuation-driven firing rate dynamics". Basically, I didn't understand what the author is trying to achieve. Clearly it is not some chaotic dynamics a la Sompolinsky-Crisanti-Sommers (though this is one of the references cited), because this has already been demonstrated in Fig. 1. Is it some meta-stability as in refs 23 and 24? If so, this should be clearly stated, possibly citing the relevant references. Also, if so, it seems that the proposed solution does not solve the basic problem of refs 23-24; that is, metastability is essentially a finite-size effect (here, controlled by the parameter K_\sigma). Please clarify these points.

In the same section, lines 418-426. This is confusing. In the network, synaptic interactions (i.e., the J's of Eq. (4)) are both negative and positive, though their variances are obviously positive. Clearly, you must be talking about the "interactions" among the mean-field populations (i.e., Eqs. (5), (9) and (10)). See also my previous comment.

The author carefully discusses the possible biological mechanisms underlying the WTA mechanism. It is not clear, however, that any of these mechanisms (or even a combination of them) would lead to a "perfect" WTA as the one used in the theory and simulations. As it is implemented now the mechanism is 'fiercely' non-local (despite the author calling it 'local' several times); the activity of a neuron depend on the same-time inputs to other neurons, i.e., it is non-local in time. So, it seems to me, there is also an issue of time scales that would require a short discussion; the time scale over which the WTA is achieved vs. the time scale of the synaptic interaction between the different units.

As an historical aside, the same mechanism was considered by Bienestock ('A model of cortex', 1995) with motivations partially overlapping with the ones behind the present work. In particular, he introduces a dynamics similar to the one studied here and faces more or less the same problems in attempting a 'biological justification'.

Finally, perhaps I'm too old and my eyesight is not as good as it used to be, but figures are really small. A simple solution would be to reduce the number of sub-figures within a 'figure'. Obviously, the author is free to ignore this suggestion.

Reviewer #3: In this paper, Pang analyzes a network of spiking neurons incorporating a winner-take-all mechanism. By studying the network dynamics in the mean-field limit, the work demonstrates that such networks can exhibit a fluctuation-driven regime supporting flexible, nonlinear firing rate dynamics. These findings are further substantiated through simulations of the microscopic model. The observed behavior is attributed to the modulation of synaptic weight variance between distinct neuronal blocks.

The study is well-written, the mathematical derivations are robust, and the biological plausibility of the mechanism is appropriately discussed in the manuscript.

While alternative mechanisms capable of generating nonlinear rate dynamics exist and the model’s units are not biologically realistic - as acknowledged by the author - I believe this work remains valuable and relevant to the research community.

My main concern lies with the manuscript's mention of "computational" properties. While I agree that nonlinear dynamics, multistability, and sequential activation are fundamental building blocks of computational capabilities, no explicit computations are presented in this manuscript. Although it is reasonable to expect that the network could perform tasks similar to other rate models, this is not explicitly shown here. As such, I suggest framing the references to computational properties in a more conditional manner, for example, by emphasizing the potential for such applications rather than asserting them outright, particularly in the abstract (second-to-last paragraph) and the author summary (last two paragraphs).

Other suggestions:

- An alternative mechanism for the generation of nonlinear dynamics in balanced networks has been proposed by Sanzeni, Histed, and Brunel (2020) in Response nonlinearities in networks of spiking neurons (PLoS Computational Biology). They show that while a linear transfer function is obtained in the strong coupling limit, nonlinearities emerge at response-onset and saturation as the coupling strength decreases. The author might consider discussing why a winner-take-all mechanism is necessary instead of this approach (maybe related to the finite-size theories mentioned in line 27?)

- line 488-489: The author might consider comparing the proposed approach to the work of Recanatesi et al. (2022) in Metastable attractors explain the variable timing of stable behavioral action sequences (Neuron). In this study, transitions between attractors are driven by low-dimensional correlated variability in the input, which could provide a useful perspective for generating more complex temporal dynamics in balanced networks.

- Another important point concerns the figure captions, which are somewhat difficult to read due to their small size. This may be related to the images not spanning the full page width. Enhancing the caption size and ensuring figures are appropriately scaled would improve the manuscript’s clarity.

**Have the authors made all data and (if applicable) computational code underlying the findings in their manuscript fully available?**

Reviewer #1: Yes

Reviewer #2: None

Reviewer #3: Yes

PLOS authors have the option to publish the peer review history of their article (what does this mean?). If published, this will include your full peer review and any attached files.

Reviewer #1: No

Reviewer #2: No

Reviewer #3: No

**Figure resubmission:**
---

## [Decision Letter · Decision Letter 1]

Dear Mr. Pang,

We are pleased to inform you that your manuscript 'Balanced state of networks of winner-take-all units' has been provisionally accepted for publication in PLOS Computational Biology.

Best regards,

Jonathan Rubin

Academic Editor

PLOS Computational Biology

Daniele Marinazzo

Section Editor

PLOS Computational Biology

Reviewer's Responses to Questions

**Comments to the Authors:**

Reviewer #1: The authors addressed all of my concerns.

Reviewer #2: The author addressed all my minor concerns satisfactorily.

Reviewer #3: Thank you for addressing the concerns raised in the previous round of review. The revisions have been thorough and have improved the clarity and quality of the manuscript. In its current form, the paper is suitable for publication.

**Have the authors made all data and (if applicable) computational code underlying the findings in their manuscript fully available?**

Reviewer #1: Yes

Reviewer #2: Yes

Reviewer #3: Yes

PLOS authors have the option to publish the peer review history of their article (what does this mean?). If published, this will include your full peer review and any attached files.

Reviewer #1: **Yes: **Robert Rosenbaum

Reviewer #2: No

Reviewer #3: No

---

## [Editor Report · Acceptance letter]

PCOMPBIOL-D-24-02146R1

Balanced state of networks of winner-take-all units

Dear Dr Pang,

I am pleased to inform you that your manuscript has been formally accepted for publication in PLOS Computational Biology. Your manuscript is now with our production department and you will be notified of the publication date in due course.

With kind regards,

Anita Estes
